# Towards Codable Watermarking for Injecting Multi-Bits Information to LLMs

**Lean Wang**[*,†,§]**, Wenkai Yang**[*,‡,§]**, Deli Chen**[*,¶]**,**
**Hao Zhou**[§]**, Yankai Lin**[‡]**, Fandong Meng**[§]**, Jie Zhou**[§]**, Xu Sun**[†]

[†]National Key Laboratory for Multimedia Information Processing,
 School of Computer Science, Peking University
[‡]Gaoling School of Artificial Intelligence, Renmin University of China
[§]Pattern Recognition Center, WeChat AI, Tencent Inc., China
[¶]DeepSeek AI
`{lean, xusun}@pku.edu.cn, victorchen@deepseek.com`
`{wenkaiyang, yankailin}@ruc.edu.cn`
`{tuxzhou, fandongmeng, withtomzhou}@tencent.com`

## Abstract

As large language models (LLMs) generate texts with increasing fluency and realism, there is a growing need to identify the source of texts to prevent the abuse of LLMs. Text watermarking techniques have proven reliable in distinguishing whether a text is generated by LLMs by injecting hidden patterns. However, we argue that existing LLM watermarking methods are encoding-inefficient and cannot flexibly meet the diverse information encoding needs (such as encoding model version, generation time, user id, etc.). In this work, we conduct the first systematic study on the topic of **Codable Text Watermarking for LLMs** (CTWL) that allows text watermarks to carry multi-bit customizable information. First of all, we study the taxonomy of LLM watermarking technologies and give a mathematical formulation for CTWL. Additionally, we provide a comprehensive evaluation system for CTWL: (1) watermarking success rate, (2) robustness against various corruptions, (3) coding rate of payload information, (4) encoding and decoding efficiency, (5) impacts on the quality of the generated text. To meet the requirements of these non-Pareto-improving metrics, we follow the most prominent vocabulary partition-based watermarking direction, and devise an advanced CTWL method named **Balance-Marking**. The core idea of our method is to use a proxy language model to split the vocabulary into probability-balanced parts, thereby effectively maintaining the quality of the watermarked text. Extensive experimental results show that our method outperforms the baseline under comprehensive evaluation. Our code is available at `https://github.com/lancopku/codable-watermarking-for-llm`.

## 1 Introduction

Recently, with the explosive development of Large Language Models (**LLMs**) (OpenAI, 2022; Touvron et al., 2023), there has been growing concern in the community about the potential negative effects of the AI-generated content (**AIGC**). For instance, LLMs could be exploited to produce fake news, encyclopedia entries, or academic papers. Hence, there is an urgent need to reliably distinguish between human-written and AI-generated texts.

Text watermarking (Jalil & Mirza, 2009; Venugopal et al., 2011) aims to inject hidden patterns into the generated text, and detect the specific patterns to determine the source of text. The most representative line of LLM watermarking methods (Kirchen. et al., 2023a; Lee et al., 2023; Zhao et al., 2023) injects the watermark by controlling the available part of the vocabulary during LLM's decoding process and can detect whether a text contains a watermark with high accuracy and low false positive rate. However, we argue that these existing LLM watermarking methods encode too

---

[*]   Equal contribution.

Table 1: Taxonomy of LLM watermarking technologies, with a representative work of each direction attached. It can be found that existing LLM watermarking methods either do not make full use of the generation ability of LLMs, or lack customized watermarking information. Our work simultaneously addresses both of these issues, filling the gap in this line of academic research.

| Watermark Information | Watermark Injection Timing | |
|---|---|---|
| | Post-process after LLM Generation | Integrate with LLM Generation |
| One-Bit | Black-Box Watermarking (Yang et al., 2023b) | LLM Watermarking (Kirchen. et al., 2023a) |
| Multi-Bits | Natual Language Watermarking (Yoo et al., 2023a) | Codable Text Watermarking for LLMs (This Work) |

limited information (only 1 bit of information - whether the text is generated by one specific model or not), and can not satisfy the increasing demand for customizing information in the application of LLMs (for example, embedding model and version information in the watermark can effectively trace the source of a text among multiple LLMs, etc.).

**Present Work.** We conducted the first systematic study on the topic of Codable Text Watermarking for LLMs (**CTWL**), which allows the watermark injected into LLM-generated text to carry more customizable information. First and foremost, given the fact that the boundary and definition of CTWL remain unclear, we start with a taxonomy study on watermarking technologies (Section 2) and then give a mathematical formalization of the CTWL problem (Section 3). Then, we propose a comprehensive evaluation system for CTWL (Appendix C), which consists of the following 5 criteria: (1) **Success Rate**: the LLM watermarking method should have high success rates of recovering message from watermarked texts and distinguishing between human-written and watermarked ones. (2) **Robustness**: the watermarking method should remain a high success rate when facing different challenging attacks (such as copy-paste attack, substitution attack, etc.). (3) **Coding Rate**: the ratio of text token number to information bit number should be high. (4) **Computational Complexity**: the computation cost of watermark writing and decoding should meet the practical hardware and latency requirements. (5) **Text Quality**: the impact of adding complex watermark patterns on the quality of generated texts should be minimal.

Therefore, devising a practically applicable CTWL method is a challenging task, since it is difficult to make Pareto improvements in these conflicting metrics. In this work, we first extend the random vocabulary splitting idea from Kirchen. et al. (2023a), and devise the **Vanilla-Marking** method to encode a multi-bit message in watermark. However, this naive baseline perform poorly in maintaining generated text quality, since multi-bit watermark is more complex than one-bit watermark, thus injecting multi-bit watermark causes a greater impact on the text quality.

To overcome this challenge, and inspired by the idea that a probability-balanced vocabulary partition can more effectively ensure the quality of the watermarked text, we devise an advanced CTWL method named **Balance-Marking**. Specifically, we leverage a proxy language model (**proxy-LM**) to guide the vocabulary partitioning instead of random splitting, ensuring that the probabilities of the available/unavailable parts are close to 50-50. This balance partition avoids the excessive impact of watermark on text quality caused by random vocabulary splitting(e.g. most high-probability tokens are in the unavailable part). Also, this mechanism implicitly helps the watermark algorithm to skip low-entropy segments of text, which further ensures the text quality. Moreover, we can achieve a flexible and customizable watermarking algorithm deployment for different LLM application scenes by switching the proxy-LM. For example, we can take a small language model as the proxy-LM to ensure a higher inference efficiency; we can take a public language model as the proxy-LM and achieve an open watermarking protocol among different LLM providers without the leakage of the LLMs. [1] Extensive experimental results demonstrate that the Balance-Marking method surpasses the Vanilla-Marking method across different sizes of LLMs (OPT-1.3B (Zhang et al., 2022), LLaMA-7B and LLaMA-13B (Touvron et al., 2023)). Additional experiments are performed to assess the influence of crucial modules within our approach. Furthermore, we analyze the application scenarios of CTWL and potential avenues for future research.

---

[1]A more detailed discussion on the application scenarios of codable text watermark and the selection of proxy language models is presented in the Appendix D.

## 2 RELATED WORK

The current studies on identifying LLM-generated texts can be mainly divided into two categories: detecting-based (introduced in Appendix A) and watermarking-based methods. The watermarking-based method has proven more effective and reliable Sadasivan et al. (2023) and we focus on the watermarking-based method. Text watermarking technology marks text by injecting human-invisible hidden patterns into the text, and then determines the text source by detecting whether the text contains a watermark (Jalil & Mirza, 2009; Venugopal et al., 2011; Abdelnabi & Fritz, 2021; Yang et al., 2022). We analyze the taxonomy of LLM watermarking methods in Table 1. LLM watermarking techniques can be categorized into two types according to the point of watermark insertion: (1) Integrated into the model generation process (Kirchen. et al., 2023a), and (2) Post-processing after text generation (Yoo et al., 2023a; Yang et al., 2023b). Post-processing methods are independent of LLMs and they usually utilize masked language models (MLMs) (e.g. BERT (Devlin et al., 2019), RoBERTa (Liu et al., 2019)) to replace tokens with synonyms. Both one-bit Yang et al. (2023b) or multi-bits Yoo et al. (2023a) information can be injected in this manner. However, we argue that post-processing methods are not as effective as integrating methods for generating high-quality watermarked texts, since they do not take advantage of the generative power of LLMs. Moreover, replacement-based methods can only operate individual tokens and cannot adaptively change the subsequent generation sequence like integrating methods.

The most representative integrating watermarking technique for LLMs is proposed by Kirchen. et al. (2023a), which watermarks the LLM-generated text by manipulating the available part of the vocabulary. Following Kirchen. et al. (2023a), several studies explore effective text watermarking in various scenarios. Kirchen. et al. (2023b) later improves this method by exploring diverse choices of random seed generators for splitting the vocabulary. Lee et al. (2023) takes a step further to actively select high entropy tokens to watermark instead of all tokens, which is shown to be more effective in the code detection task. Zhao et al. (2023) simplify the watermark method of Kirchen. et al. (2023a) by making the vocabulary splitting independent of the previously generated tokens and only dependent on a global key. However, all these methods generate watermarks that only contain 1 bit of information, which cannot meet the need to inject watermarks with diverse customized information. There are some very concurrent works (Yoo et al., 2023b; Fernandez et al., 2023) that focus on this problem. These methods still follow the random vocabulary partition (Kirchen. et al., 2023a), posing an unignorable threat to the quality of generated text caused by the multi-bits watermark. In contrast, our Balance-Marking method can effectively maintain the text quality after injecting complex messages by balancing the probability distribution of the vocabulary partitions.

## 3 MATHEMATICAL FORMULATION OF CODABLE LLM WATERMARKING

### 3.1 NOTATIONS AND PRELIMINARIES

Here, we first introduce the necessary notations used in this paper. Assume there is a large language model $LLM$ that takes a prompt sequence $\mathbf{x}^{prompt}$ as the input and sequentially outputs the corresponding tokens to form natural sentences as the response. At $l$-th step ($l = 1, 2, \cdots, L$), the entire input for $LLM$ is the combination of the original prompt $\mathbf{x}^{prompt}$ and the sequence of tokens $\mathbf{t}_{:(l-1)} = \{t_0, \cdots, t_{l-1}\}$ that is already predicted by $LLM$ in previous steps.[2] Then, $LLM$ produces a probability distribution over the entire vocabulary $\mathcal{V}$ as $\mathbf{P}_{LLM}(\mathbf{x}^{prompt}, \mathbf{t}_{:(l-1)}) = (\cdots, P_{LLM}(v|\mathbf{x}^{prompt}, \mathbf{t}_{:(l-1)}), \cdots)$, in which $P_{LLM}(v|\mathbf{x}^{prompt}, \mathbf{t}_{:(l-1)})$ is the predicted probability of a specific token $v$ by $LLM$. The next token $t_l$ is sampled based on $\mathbf{P}_{LLM}(\mathbf{x}^{prompt}, \mathbf{t}_{:(l-1)})$ according to specific sampling rules, such as multinomial sampling or greedy sampling.

### 3.2 FORMULATION OF CODABLE TEXT WATERMARKING

In this paper, we propose the concept of **codable text watermarking for LLMs** (**CTWL**) that can encode rich and necessary information into the text watermarks, in order to satisfy the multiple demands on the realistic application of LLMs. Formally, CTWL can be formulated as a *message*

---

[2]In the first step, there is no previously generated token, thus define $t_0$ is none.

*encoding* stage with a *message decoding/extracting* stage:

$$\text{Encoding:} \quad \mathcal{P} \times \mathcal{M} \to \mathcal{T}, \quad Enc(\mathbf{x}^{prompt}, m) = \mathbf{t},$$
$$\text{Decoding:} \quad \mathcal{T} \to \mathcal{M}, \quad Dec(\mathbf{t}) = m, \tag{1}$$

where $\mathcal{P}$, $\mathcal{T}$ and $\mathcal{M}$ represents the prompt space, text space, and message space separately, $m \in \mathcal{M}$ is the message that needs to be encoded. Thus, one-bit watermarking methods can be considered as a simplified case of CTWL, where the message space only contains $\{0, 1\}$. Following the definition in Eq. (1), the target on decoding messages from $\mathbf{t}$ can be written as

$$m = \arg\max_{m' \in \mathcal{M}} P_w(m'|\mathbf{t}), \tag{2}$$

where we need to design a specific probability function $P_w$ referred to as **message function** (refer to Section 4.2.) to measure how likely is that the watermarked message is $m'$ given $\mathbf{t}$.

Therefore, according to Bayes Formula, in the message encoding phase, it is equivalent to achieve[3]

$$\max_{\mathbf{t}} \{P_w(\mathbf{t}|m) / \max_{m' \neq m} P_w(\mathbf{t}|m')\}$$
$$\Longleftrightarrow \max_{\mathbf{t}} \{\sum_{l=1}^{L} \log P_w(t_l|m, \mathbf{t}_{:(l-1)}) - \max_{m' \neq m} \sum_{l=1}^{L} \log P_w(t_l|m', \mathbf{t}_{:(l-1)})\}. \tag{3}$$

That is, we aim to enlarge the gap between the probability that the text $\mathbf{t}$ is generated under message $m$ and the probability that it is generated under other $m'$.[4] However, as we can see, the above equation only considers the target of effectively hiding message $m$ into $\mathbf{t}$, but does not take the quality of the generated text into consideration. Thus, we take the original text generated without embedded watermarks $\mathbf{t}^{ori}$ as a baseline for comparison, and reformulate the encoding phase as:

$$\max_{\mathbf{t}} \{\sum_{l=1}^{L} \log P_w(t_l|m, \mathbf{t}_{:(l-1)}) - \max_{m' \neq m} \sum_{l=1}^{L} \log P_w(t_l|m', \mathbf{t}_{:(l-1)})\},$$
$$\text{s.t.} \quad \text{PPL}(\mathbf{t}|\mathbf{x}^{prompt}) \leq \text{PPL}(\mathbf{t}^{ori}|\mathbf{x}^{prompt}) + \epsilon. \tag{4}$$

Here, we utilize the perplexity (PPL) metric to measure the quality of the generated text, with the aim of ensuring the watermarked text maintains a similar quality to the text without watermarks.

## 4 BALANCE-MARKING: A SIMPLE YET EFFECTIVE CTWL METHOD

In this section, we first present an approximation algorithm to solve Eq. (4) in Section 4.1, which serves as a general encoding algorithm for any predefined $P_w$. Then, we introduce two designs of $P_w$ in Section 4.2 as unified guidelines to encode and decode messages. Finally, given the pre-defined encoding algorithm and $P_w$, the message extracting can be performed as presented in Section 4.3.

### 4.1 A GENERAL FRAMEWORK FOR CODABLE WATERMARK ENCODING

In order to solve the constrained optimization problem in Eq. (4), we are motivated to apply the method of Lagrange Multipliers by introducing a dual variable $\lambda$ and turn it into an unconstrained optimization problem. Furthermore, assume that the PPL scores in Eq. (4) are calculated based on the same $LLM$ used to generate $t$,[5] given the prompt $\mathbf{x}^{prompt}$ and $LLM$, the term $\text{PPL}_{LLM}(\mathbf{t}^{ori}|\mathbf{x}^{prompt})$ can be regarded as a constant. Therefore, the target can be rephrased as:[3]

$$\max_{\mathbf{t}} \sum_{l=1}^{L} \{\log P_{LLM}(t_l|\mathbf{x}^{prompt}, \mathbf{t}_{:(l-1)}) + \delta(\log P_w(t_l|m, \mathbf{t}_{:(l-1)}) - \log P_w(t_l|\hat{m}, \mathbf{t}_{:(l-1)})\},$$

$$\tag{5}$$

where we use $\text{PPL}(\mathbf{t}|\mathbf{x}^{prompt}) = [\Pi_{l=1}^{L} P_{LLM}(t_l|\mathbf{x}^{prompt}, \mathbf{t}_{:(l-1)})]^{-\frac{1}{L}}$, and set $\delta = \frac{L}{\lambda}$ and $\hat{m} = \text{argmax}_{m' \neq m} \sum_{l=1}^{L} \log P(t_l|m', t_{:(l-1)})$.

---

[3]The detailed derivations can be found in Appendix B.

[4]We take the form of division in Eq. (3) in order to successfully derive the optimization target to Eq. (5).

[5]This assumption is practical as the LLM is powerful enough to accurately measure the perplexity of a text.

---

**Algorithm 1:** A General Message Encoding Framework for A Settled $P_w$

---

**Input:** Language model $LLM$, prompt $\mathbf{x}^{prompt}$, message $m$, watermarking weight $\delta$

**for** $l = 1, \cdots, L$ **do**

    1. Calculate $\log P_{LLM}(v|\mathbf{x}^{prompt}, \mathbf{t}_{:(l-1)})\}$ for each $v$ in the vocabulary using $LLM$;

    2. Calculate $\log P_w(v|m, \mathbf{t}_{:(l-1)})$ based on the settled $P_w$;

    3. Select $t_l = \arg\max_{v} \{\log P_{LLM}(v|\mathbf{x}^{prompt}, \mathbf{t}_{:(l-1)}) + \delta(\log P_w(v|m, \mathbf{t}_{:(l-1)}) -$

    $\frac{1}{|\mathcal{M}|} \sum_{m' \in \mathcal{M}} \log P_w(v|m', \mathbf{t}_{:(l-1)})\}$

**end**

**Output:** watermarked text $\mathbf{t} = \{t_1, t_2, \cdots, t_L\}$

---

Eq. (5) motivates us that in order to encode $m$ into $\mathbf{t}$, we can manipulate the output logits during each token's generation by adding a term $\delta(\log P_w(t_l|m, \mathbf{t}_{:(l-1)}) - \log P_w(t_l|\hat{m}, \mathbf{t}_{:(l-1)}))$ to the original log logits. **However, in practice, solving $\hat{m}$ is infeasible because the true $\hat{m}$ can only be solved after the whole output $t$ is determined, while we need to calculate** $\delta(\log P_w(t_l|m, \mathbf{t}_{:(l-1)}) - \log P_w(t_l|\hat{m}, \mathbf{t}_{:(l-1)}))$ **in each generation step according to Eq. (5).** Therefore, we replace $\log P_w(t_l|\hat{m}, \mathbf{t}_{:(l-1)})$ with calculable $\frac{1}{|\mathcal{M}|} \sum_{m' \in \mathcal{M}} \log P_w(t_l|m', \mathbf{t}_{:(l-1)})$ as an alternative, and finally get the message encoding object function in each generation step as:

$$L(m, \mathbf{x}^{prompt}, \mathbf{t}_{:(l-1)}) = \max_{v}\{\underbrace{\log P_{LLM}(v|\mathbf{x}^{prompt}, \mathbf{t}_{:(l-1)})}_{\text{model logit}}$$

$$+ \underbrace{\delta(\log P_w(v|m, \mathbf{t}_{:(l-1)}) - \frac{1}{|\mathcal{M}|} \sum_{m' \in \mathcal{M}} \log P_w(v|m', \mathbf{t}_{:(l-1)}))}_{\text{message logit}}\}, \tag{6}$$

where we denote the first term in the right as the **model logit**, which is determined by the $LLM$ only; we denote the second additional term as the **message logit**, which is the key component for encoding message $m$ into $t$.

As we can see, as long as the function $P_w$ is well-defined, the encoding process can be completed by adding the message logits to the model logits and sampling the token based on the new logits. We put the general message encoding procedure in Algorithm 1,[6] and will discuss how to properly design $P_w$ in detail in the next section.

### 4.2 THE DESIGN OF MESSAGE FUNCTION $P_w$

In the above section, we present a general encoding algorithm for an arbitrary $P_w$. In the following, we will introduce two designs of $P_w$ as our preliminary attempts toward CTWL.

#### 4.2.1 VANILLA $P_w$ FOR RANDOM VOCABULARY PARTITION

According to Eq. (6), a high value of $L(m, \mathbf{x}^{prompt}, \mathbf{t}_{:(l-1)})$ relies on the existence of a $v$ with a high message logit. In other words, there should be a $v$ for which $\log P_w(v|m, \mathbf{t}_{:(l-1)})$ greatly surpasses the mean. To achieve this, one natural idea is to ensure that the distribution $\mathbf{P}_w(m, \mathbf{t}_{:(l-1)}) = (P_w(v_1|m, \mathbf{t}_{:(l-1)}), P_w(v_2|m, \mathbf{t}_{:(l-1)}), \cdots, P_w(v_{|\mathcal{V}|}|m, \mathbf{t}))$ varies greatly across distinct messages. In this way, for a message $m$, there would at least exist a $v$ whose $\log P_w(v|m, \mathbf{t}_{:(l-1)})$ deviates far from the mean $\frac{1}{|\mathcal{M}|} \sum_{m' \in \mathcal{M}} \log P_w(v|m', \mathbf{t}_{:(l-1)})$, thereby resulting in a high message logit. To ensure such differences in $\mathbf{P}_w(m, \mathbf{t}_{:(l-1)})$ with different $m$, we can assign random values for $\mathbf{P}_w(m, \mathbf{t}_{:(l-1)})$ based on the random seeds directly decided by their own $m$:

$$\log \hat{P}_w(v|m, \mathbf{t}_{:(l-1)}) = \begin{cases} 1, & h(v, m, \mathbf{t}_{:(l-1)}) = 1, \\ 0, & h(v, m, \mathbf{t}_{:(l-1)}) = 0. \end{cases} \tag{7}$$

---

[6]Though we employ a greedy search as the text generation algorithm in Algorithm 1 for example, our framework is also compatible with other generation rules such as beam search.

---

**Algorithm 2:** Algorithm of Choosing Subset $V_{m,\mathbf{t}_{:(l-1)}}$ (Practical Version in Red)

---

**Input:** Message $m$, text prefix $\mathbf{t}_{:(l-1)}$, language model $LLM$, proxy-LM $LM_{proxy}$,
$\qquad \mathcal{M}_\mathcal{A} = \{1, \cdots, A\}$, hash functions $h$ and $\hat{h}$.
1. Calculate a seed $s = h(m, \mathbf{t}_{:(l-1)})$, or $s = h(\hat{h}(m), \mathbf{t}_{:(l-1)})$ where $\hat{h}$ maps $m$ to
$\quad \hat{h}(m) \in \mathcal{M}_A$;
2. Shuffle the vocab list $(v_1, \cdots, v_{|\mathcal{V}|})$ to $(v'_1, \cdots, v'_{|\mathcal{V}|})$ with the seed $s$;
3. Select the first $k$ tokens in the shuffled list so that $k$ is the minimal value to make
$\quad \{v'_1, \cdots, v'_k\}$ satisfy Eq. (9) or Eq. (11).
**Output:** $V_{m,\mathbf{t}_{:(l-1)}} = \{v'_1, \cdots, v'_k\}$

---

$$\log P_w(v|m, \mathbf{t}_{:(l-1)}) = \log \frac{\hat{P}_w(v|m, \mathbf{t}_{:(l-1)})}{\sum\limits_{v} \hat{P}_w(v|m, \mathbf{t}_{:(l-1)})}. \tag{8}$$

In the above, $h$ denotes a hash function that maps the input $(v, m, \mathbf{t}_{:(l-1)})$ to either 0 or 1. This can be considered as a vanilla extension from the soft watermarking method in Kirchen. et al. (2023a) by further taking $m$ into consideration, thus we denote it as **Vanilla-Marking**.

### 4.2.2 $LM_{proxy}$-AIDED $P_w$ FOR BALANCE VOCABULARY PARTITION

The message function $P_w$ proposed in Section 4.2.1 has a problem that it does not guarantee that the same $v$ can also have a high model logit at the same time. This could potentially result in a small sum of the model logit and the message logit. Therefore, we argue that **a more advanced $P_w$ should produce a $v$ with both a high model logit and a message logit**.

To accomplish this, we are motivated to utilize the model logit distribution $\mathbf{P}_{LLM}(\mathbf{x}^{prompt}, \mathbf{t}_{:(l-1)})$ as prior knowledge, and pre-select a subset of tokens that is likely to contain some tokens with high model logits in advance. Then, we assign high message logits to the tokens in the above subset, ensuring the existence of a token $v$ with both a high model logit and message logit. Formally, we propose Algorithm 2 to randomly choose the subset $V_{m,\mathbf{t}_{:(l-1)}}$ that satisfies the following condition:

$$\sum_{v \in V_{m,\mathbf{t}_{:(l-1)}}} P_{LLM}(v|\mathbf{x}^{prompt}, \mathbf{t}_{:(l-1)}) \geq \sigma, \tag{9}$$

where $\sigma$ is a controllable threshold. W set $\sigma = 0.5$ in the following paper unless otherwise stated, because we believe that balancing the probability accumulations of tokens within and out of $V_{m,\mathbf{t}_{:(l-1)}}$ can achieve the maximal diversity of $V_{m,\mathbf{t}_{:(l-1)}}$ w.r.t. different $m$. The reason why we design Algorithm 2 in such a way is, **the case when all $\{P_{LLM}(v|\mathbf{x}^{prompt}, \mathbf{t}_{:(l-1)})|v \in V_{m,\mathbf{t}_{:(l-1)}}\}$ values tend to be small, yet still sum to $\sigma$, is very unlikely to occur**. Thus, there should always be some $P_{LLM}(v|\mathbf{x}^{prompt}, \mathbf{t}_{:(l-1)})$ that is relatively large to make the summation exceed the threshold. Additionally, introducing randomness in the selection process of $V_{m,\mathbf{t}_{:(l-1)}}$ can enlarge the difference in $V_{m,\mathbf{t}_{:(l-1)}}$ among different messages $m$, which plays the same role as that in Vanilla-Marking.

After getting $V_{m,\mathbf{t}_{:(l-1)}}$, we assign the message logits by following Eq.( 8) but modifying Eq. (7) as

$$\log \hat{P}_w(v|m, \mathbf{t}_{:(l-1)}) = \begin{cases} 1, & v \in V_{m,\mathbf{t}_{:(l-1)}}, \\ 0, & v \notin V_{m,\mathbf{t}_{:(l-1)}}. \end{cases} \tag{10}$$

Also, we make some specific improvements to make our method more practical in various scenarios. We summarize these strategies briefly here, and put the detailed illustrations in Appendix F: **(1)** We omit $\mathbf{x}^{prompt}$ and truncate $\mathbf{t}_{:(l-1)}$ to a fixed-length $\mathbf{t}_{(l-1-L_{prefix}):(l-1)}$ for consistency during both encoding and decoding. **(2)** To make our method applicable in various scenarios discussed in Appendix D, we broaden the $LLM$ used in $P_{LLM}(v|\mathbf{t}_{(l-1-L_{prefix}):(l-1)})$ into a general proxy model denoted as $LM_{proxy}$, and modify the condition in Eq. (9) to:

$$\sum_{v \in V_{m,\mathbf{t}_{:(l-1)}}} P_{LM_{proxy}}(v|\mathbf{t}_{(l-1-L_{prefix}):(l-1)}) \geq \sigma. \tag{11}$$

**(3)** For efficient encoding and decoding on the entire message space, we opt to first map the entire message space into a smaller space as $m \rightarrow \hat{h}(m) \in \mathcal{M}_A = \{1, \cdots, A\}$[7] by using another hash function $\hat{h}$, and then compute the seed $s$ as $s = h(\hat{h}(m), x_{:(l-1)})$.

The practical version of the above $LM_{proxy}$-aided watermarking method is specially highlighted in red in Algorithm 2. Given that it employs a probability-balanced vocabulary partition, we refer to it as **Balance-Marking**.

Here, we briefly discuss about the two advantages that Balance-Marking has compared to Vanilla-Marking. (1) First, **Balance-Marking is more effective in maintaining text quality**. That is because the way of pre-choosing the subset $V_{m,\mathbf{t}_{:(l-1)}}$ ensures that Balance-Marking can find tokens with both high model logits and message logits, thus making the next generated word more meaningful and reliable. However, Vanilla-Marking, which just randomly selects the available tokens, might end up with an unreasonable next token. (2) Secondly and interestingly, **Balance-Marking also has the ability to automatically bypass low-entropy sections of the text**, which is a property that Lee et al. (2023) explicitly aims to achieve. For example, if there is only one reasonable token candidate whose predicted probability by proxy-LM is almost 1.0, then for all messages, this token would be selected into the available part. In other words, this position in the sequence is implicitly "skipped" during watermark encoding and decoding, and the decoding process is actually carried out by comparing the values of $P_w$ under different $m$ on those high-entropy sections of the text.

### 4.3 MESSAGE DECODING

For a settled $P_w$, the decoding process is conducted by finding a solution to Eq. (2). As per Bayes' Theorem, this can be rewritten as:

$$m = \underset{m' \in \mathcal{M}}{\arg\max}\, P_w(\mathbf{t}|m'), = \underset{m' \in \mathcal{M}}{\arg\max}\{\sum_{l=1}^{L} \log P_w(t_l|m', \mathbf{t}_{:(l-1)})\}. \tag{12}$$

Eq. (12) can be computed directly using either Eq. (7) and Eq. (8) for Vanilla-Marking, or Eq. (10) and Eq. (8) for Balance-Marking. Furthermore, when we need to encode multiple messages into text, we can sequentially encode each message into one segment of the text (e.g., every 100 tokens of the text). Then, message decoding can be conducted independently within each segment.

## 5 EXPERIMENT

### 5.1 EXPERIMENTAL SETTINGS

Following the experimental settings used by Kirchen. et al. (2023a), we utilize OPT-1.3B (Zhang et al., 2022) for the generation of texts in main experiments. We also conduct experiments on LLaMA-7/13B (Touvron et al., 2023) models. Our prompt inputs are derived from the news-like subset of the C4 dataset (Raffel et al., 2019). In each experiment, we extract 500 prompt inputs and truncate them to a uniform length of 300 tokens. The language model is then requested to generate 200 tokens via a 4-way beam search. To mitigate repetition in the generated text, we implement a repetition penalty of 1.5. We evaluate the quality of the text by perplexity. For OPT-1.3B, we use OPT-2.3B to calculate perplexity. As for LLaMA-7/13B, we use LLaMA-33B to calculate perplexity. The evaluation protocol for CTWL is introduced in Appendix C. Besides, a case study is provided in Appendix K.

As for the $LM_{proxy}$ in Balance-Marking, we opt to use GPT2 (Radford et al., 2019), a well-known, publicly available, and comparatively smaller language model, which comprises 124M parameters. Additional hyper-parameters used in Balance-Marking are set to the following: $A = 100$, $L_{prefix} = 10$, $\sigma = 0.5$ and $\mathcal{M} = \{0, 1, ..., 2^{20} - 1\}$. Here, a message $m \in \mathcal{M}$ corresponds to 20-bit information. The hash scheme is the same as Kirchen. et al. (2023a). A detailed discussion is presented in G. Besides, we conduct some approximation for acceleration, which is detailed in Appendix E.

---

[7]For a detailed discussion about the choice of $A$, refer to Appendix G.2.

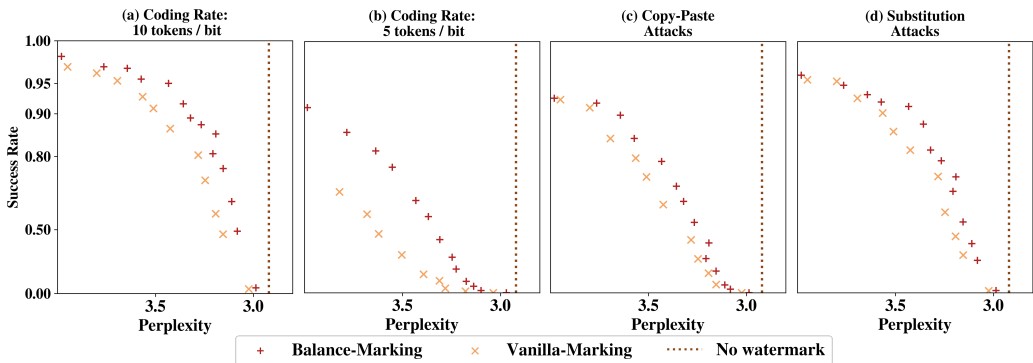

Figure 1: Balance-Marking outperforms Vanilla-marking under both different coding rates (subfigure a, b) and different attack scenarios (subfigure c, d).

## 5.2 THE RESULTS OF WATERMARK QUALITY

We experiment with Vanilla-Marking and Balance-Marking by adjusting the parameter $\delta$ in Algorithm 1. This parameter helps strike a balance between text quality and watermark success rate. Higher $\delta$ values lead to a stronger watermark but at the expense of text quality. We run tests using a range of $\delta$ values (i.e., $\{0.5, 0.8, 0.9, 1.0, 1.1, 1.2, 1.3, 1.4, 1.5, 1.8, 2.0, 2.5, 3.0, 4.0\}$) and coding rates of 10 or 5 tokens per bit to observe the corresponding effects.

Figure 1a and 1b depict the trade-off between text quality, assessed by perplexity, and the success rate of recovering the embedded watermark. Enhancing the watermark's success rate tends to adversely affect text quality, and embedding watermarks with a high coding rate of 5 tokens per bit is particularly challenging. In both cases, **Balance-Marking outperforms Vanilla-Marking, offering a superior trade-off between text quality and watermark success rate.**

## 5.3 THE RESULTS OF ROBUSTNESS TO REAL-WORLD ATTACKS

In real-world applications, the embedded watermark in a text can be weakened by different types of attacks. For example, watermarks can be difficult to detect when hidden in human-written texts, a situation known as Copy-Paste Attacks (Kirchen. et al., 2023b). Moreover, the watermark can be eroded by actions like word substitution, referred to as Substitution Attacks. In our study, we apply these attacks to the watermark under a coding rate of 10 tokens per bit.[8]

**Robustness to Copy-paste Attacks.** A possible way to use LLM-generated text is to integrate it into human-written documents. We refer to this as a "Copy-Paste Attack" (Kirchen. et al., 2023b). This presents a challenge in watermark detection as the location of the watermarked text becomes unpredictable. To simulate this, we insert a 200-token watermarked text into a 1000-token piece of human-written text, taken from the C4 dataset. The detection process involves using a sliding window technique (Kirchen. et al., 2023b) and computing $P_w(m|\mathbf{x}_{sliding\_window})$.[9] To avoid incorrectly labeling human-written texts as watermarked texts, we set a threshold of $1 - 10^{-5}$. Any $\mathbf{x}_{sliding\_window}$ with $\max_m P(m|\mathbf{x})$ below this threshold will make it be classified as human-written. Our tests reveal that this threshold successfully prevents mislabeling human-written texts as watermarked. Figure 1c demonstrates the robustness of both Balance-Marking and Vanilla-Marking methods against copy-paste attacks, with Balance-Marking maintaining superior performance.

**Robustness to Substitution Attacks.** In the practical application of model-generated texts, tokens may be replaced for editing purposes or to prevent watermark detection. To replicate this, we employ the RoBERTa-Large (Liu et al., 2019) model to carry out word substitution (see details in

---

[8]We omit the case of coding rate of 5 tokens per bit since it makes both watermarks ineffective under attacks.

[9]Applying Bayes' Theorem, we know that $P_w(m|\mathbf{x})$ is proportional to $P_w(\mathbf{x}|m)$, which can be computed as explained in Section 4.3. By normalizing $P_w(\mathbf{x}|m)$, we obtain $P_w(m|\mathbf{x})$.

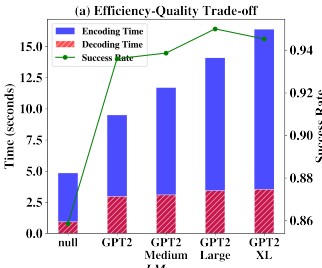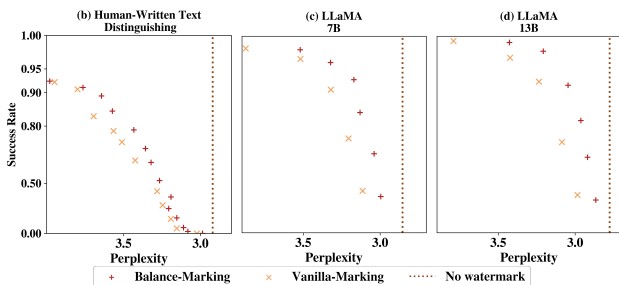

Figure 2: (a) Trade-off between efficiency and watermark success rate. Specifically, Vanilla-Marking can be viewed as Balance-Marking with $LM_{proxy} = \varnothing$ (denoted as "null"). (b) Balance-Marking outperforms Vanilla-Marking in distinguishing between message-embedded and human-written texts. (c, d) Balance-Marking is still superior to Vanilla-Marking when using LLaMA-7/13B.

Appendix H). Figure 1d illustrates the impact of Substitution Attacks (substitution ratio=5%) on Vanilla-Marking and Balance-Marking. Balance-Marking outperforms Vanilla-Marking as under Copy-Paste Attacks. Experiments with a substitution ratio of 10% are reported in Appendix I.

## 5.4 FURTHER ANALYSIS

**Distinguishing between message-embedded and human-written texts.** In practice, it's very important to ensure that human-written texts are not wrongly identified as watermarked. To prevent extracting pseudo messages from human-written texts, we use the same threshold $1 - 10^{-5}$ as in Section 5.3. While this threshold guarantees to prevent all human-written texts from being incorrectly identified, it may also misidentify some watermarked texts as human-written. Figure 2b demonstrates this by showing both the success rate of correctly distinguishing message-embedded texts from human-written ones, and the corresponding text quality (the coding rate is 10 tokens per bit). Still, Balance-Marking outperforms Vanilla-Marking.

**Trade-off between Efficiency and Watermark Quality.** The $LM_{proxy}$ introduced in Balance-Marking brings better watermark quality but also along with extra computational cost. In Figure 2a, we verify this trade-off between the efficiency and watermark quality when larger $LM_{proxy}$s are introduced. Specifically, we can view Vanilla-Marking as a special kind of Balance-Marking with $LM_{proxy} = \varnothing$. We also discussed the $LM_{proxy} = LLM$ case in Appendix G.1

**Scaling to larger LLMs.** To verify that our watermark algorithm also works for larger LLMs, we conduct experiments with LLaMA-7/13B (under a coding rate of 10, and a smaller $\delta$ set $\{0.8, 1.0, 1.2, 1.5, 2.0, 3.0\}$ to save computational costs), as shown in Figure 2d. Appendix J also lists the results with a coding rate of 5. Balance-Marking stably outperforms Vanilla-Marking when scaling to larger LLMs.

## 6 CONCLUSION

In this work, we provide the first systematic study on the topic of Codable Text Watermarking for Large Language Models (**CTWL**), filling the research gap of integrating multi-bit information watermarks into the generation process of LLMs. We first conduct a taxonomic analysis of CTWL and provide a rigorous mathematical formalization for CTWL. We then design a CTWL method Balance-Marking, which effectively ensures the balance of the probabilities of available and unavailable vocabularies with the help of a proxy language model, thereby ensuring the high quality and diversity of the generated text. Extensive experiments have shown that our method significantly outperforms the direct baseline method in the comprehensive evaluation of five dimensions, reaching a practical level of usability. We hope that our work can help the community better understand the CTWL issue and inspire more outstanding research in the future.

ETHICAL STATEMENT

Our paper aims to provide an effective way to reduce the ethical problems brought by the LLMs. Our method can inject a watermark that carries rich information into the text generated by the LLMs, thus helping the public to identify whether the specific text is machine-generated under some necessary circumstances. For example, a machine-generated text with watermarks that carry the information of the name of the source LLM can be easily traced from the source once it is used for harmful purposes, such as creating fake news or cheating on academic writings. Therefore, our work does not have any negative ethical concerns.

REPRODUCIBILITY STATEMENT

We have provided the necessary explanations for all the assumptions we made after they are proposed in the main paper. We provide detailed mathematical derivations for the complicated formulas such as deriving Eq. (4) to Eq. (5) in Appendix B. We present the detailed illustrations of the approximations we made during mathematical derivations in Appendix E. We give the necessary illustration of the experiment settings in Section 5.1 and Section 5.1 in the main paper. The detailed explanations and explorations of training hyper-parameters are in Appendix G. The descriptions of the implementations of substitution attacks are in Appendix H.

ACKNOWLEDGEMENTS

We sincerely thank all the anonymous reviewers for their helpful suggestions on improving our manuscript. This work is supported in part by a Tencent Research Grant and the National Natural Science Foundation of China (No. 62176002). Yankai Lin and Xu Sun are the corresponding authors of this paper.

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

## A  SUPPLEMENT TO RELATED WORK

**Detecting-based Methods.**   This kind of method aims to detect whether a text is generated by a language model by formulating the task as a binary classification problem (Bakhtin et al., 2019; Jawahar et al., 2020; OpenAI, 2019). (1) One way to achieve this goal is to first collect outputs from a specific LLM along with human-written texts, and then use them to train a binary classifier Mitrović et al. (2023). For example, OpenAI (2019) fine-tune a RoBERTa classification model to detect whether a text is generated by GPT-2 model or not. Later, Mitrović et al. (2023) collect data produced by ChatGPT OpenAI (2022) and fine-tune a DistilBERT model (Sanh et al., 2019) to distinguish texts written by humans and that generated by ChatGPT. OpenAI (2023) create a more advanced classifier that can detect texts generated by a variety of diverse LLMs. (2) Besides explicitly training deep classifiers, some other methods achieve detection by exploring the statistical differences between LLM-generated texts and human-written texts (Solaiman et al., 2019; Wu et al., 2023). GPTZero (Tian, 2023) is a popular tool to identify machine-generated texts by calculating the perplexity and burstiness scores of texts and comparing them with specific thresholds. DetectGPT (Mitchell et al., 2023) claims that when a machine-generated text is perturbed, its log probability will always show a decreasing pattern, which is not the same for human-written texts. Based on this finding, it defines a log probability-based metric to help detect texts generated by LLMs. Recently, there have been some works (Koike et al., 2023; Yang et al., 2023a) exploring the detection of machine-generated text in specific domains. However, all these detection methods face the risk of becoming increasingly ineffective in the future, as the LLMs will be consistently improved to behave more and more like humans, which finally makes detection impractical (Sadasivan et al., 2023).

## B  DETAILED MATHEMATICAL DERIVATIONS OMITTED IN THE MAIN PAPER

Here, we provide detailed mathematical derivations about the formulas that are simplified in the main paper.

**Eq. (3):**

$$
\max_{\mathbf{t}}\{P_w(\mathbf{t}|m)/\max_{m'\neq m}P_w(\mathbf{t}|m')\}
$$

$$
\iff \max_{\mathbf{t}}\{\prod_{l=1}^{L}P_w(t_l|,m,\mathbf{t}_{:(l-1)})/\max_{m'\neq m}\prod_{l=1}^{L}P_w(t_l|m',\mathbf{t}_{:(l-1)})\}
$$
(13)

$$
\iff \max_{\mathbf{t}}\{\sum_{l=1}^{L}\log P_w(t_l|m,\mathbf{t}_{:(l-1)}) - \max_{m'\neq m}\sum_{l=1}^{L}\log P_w(t_l|m',\mathbf{t}_{:(l-1)})\}.
$$

**From Eq. (4) to Eq. (5):**   As $\mathrm{PPL}(\mathbf{t}|\mathbf{x}^{prompt}) = [\Pi_{l=1}^{L}P(t_l|\mathbf{x}^{prompt},\mathbf{t}_{:(l-1)})]^{-\frac{1}{L}}$, Eq. (4) is equivalent to

$$
\max_{\mathbf{t}}\{\sum_{l=1}^{L}\log P_w(t_l|m,\mathbf{t}_{:(l-1)}) - \max_{m'\neq m}\sum_{l=1}^{L}\log P_w(t_l|m',\mathbf{t}_{:(l-1)})\},
$$

$$
\text{s.t.} \quad -\frac{1}{L}[\sum_{l=1}^{L}\log P(t_l|\mathbf{x}^{prompt},\mathbf{t}_{:(l-1)})] \leq \log(\mathrm{PPL}(\mathbf{t}^{ori}|\mathbf{x}^{prompt}) + \epsilon).
$$
(14)

By applying the method of Lagrange Multipliers, we can re-define the target as:

$$
\max_{\mathbf{t}}\{\sum_{l=1}^{L}\log P_w(t_l|m,\mathbf{t}_{:(l-1)}) - \max_{m'\neq m}\sum_{l=1}^{L}\log P_w(t_l|m',\mathbf{t}_{:(l-1)})
$$

$$
-\lambda[-\frac{1}{L}[\sum_{l=1}^{L}\log P(t_l|\mathbf{x}^{prompt},\mathbf{t}_{:(l-1)})] - \log(\mathrm{PPL}(\mathbf{t}^{ori}|\mathbf{x}^{prompt}) + \epsilon)]\}.
$$
(15)

For simplicity but without loss of generality, let's assume that the PPL scores are calculated based on the same $LLM$ used to generate $t$.[10] In this case, given the prompt $\mathbf{x}^{prompt}$ and $LLM$, the term

---

[10]This assumption is practical as the LLM is powerful enough to accurately measure the perplexity of a text.

$\text{PPL}_{LLM}(\mathbf{t}^{ori}|\mathbf{x}^{prompt})$ can be regarded as a constant. Therefore, Eq. (15) can be rephrased as:

$$
\max_{\mathbf{t}} \{ \sum_{l=1}^{L} \log P_w(t_l|m, \mathbf{t}_{:(l-1)}) - \max_{m' \neq m} \sum_{l=1}^{L} \log P_w(t_l|m', \mathbf{t}_{:(l-1)})
$$
$$
+ \frac{\lambda}{L} \sum_{l=1}^{L} \log P_{LLM}(t_l|\mathbf{x}^{prompt}, \mathbf{t}_{:(l-1)}) \}. \tag{16}
$$

Let $\delta = \frac{L}{\lambda}$ and $\hat{m} = \underset{m' \neq m}{\operatorname{argmax}} \sum_{l=1}^{L} \log P(t_l|m', \mathbf{t}_{:(l-1)})$, the objective function can be restructured as Eq. (5).

## C  THE EVALUATION SYSTEM OF CODABLE LLM WATERMARKING

The prosperity of LLM technology has brought more diverse and differentiated application scenarios than traditional NLP models. Therefore, the evaluation of the practicability of LLM-generated text watermarking technology is expected to keep pace with this trend. However, existing studies lack a unified convincing evaluation system for LLM-generated text watermarking technology. In this section, we start with analyzing the uniqueness of LLM applications and establish a comprehensive evaluation system for LLM-generated text watermarking from 5 different aspects.

### C.1  WATERMARKING SUCCESS RATE

We define two indicators to measure how successful the watermark is injected into LLM-generated texts as expected: (1) Success rate of recognizing the model-generated texts from human written texts, and (2) Success rate of recovering the injected watermark message. Both of these two metrics need to be considered for CTWL; while for normal LLM watermarking (Kirchen. et al., 2023a), only the first metric need to be evaluated.

### C.2  ROBUSTNESS AGAINST ATTACKS

Texts generated by LLMs are usually modified before they are actually used for the purpose of polishing or detection escaping. Thus, the watermarking algorithms need to ensure robustness in the face of various corruptions and attacks. We summarize the two most representative attacks threatening the success of LLM watermarking: (1) **Copy-Paste Attack** (Kirchen. et al., 2023b), where LLM-generated text fragments are mixed with human-written text fragments; (2) **Substitution Attack**, where individual or sequential tokens are synonymously replaced based on human knowledge or masked language models like BERT (Devlin et al., 2019) or RoBERTa (Liu et al., 2019).

### C.3  PAYLOAD INFORMATION CODING RATE

For non-codable watermarking methods, the encoded information is always one-bit. For CTWL, we divide the number of bits carried by the watermark by the length of the covered tokens as the indicator for measuring the coding rate of the watermarking algorithm. Obviously, a good watermarking technique should encode as many bits of information as possible without sacrificing the performance on other metrics.

### C.4  ENCODING AND DECODING EFFICIENCY

Injecting watermarks during LLM's generation will inevitably increase the computational cost. Moreover, restoring multiple bits of information from the text also takes higher computational complexity than decoding 1-bit information. We argue that it is necessary to consider the additional computational complexity brought by the encoding and decoding of the large model watermarking algorithm. Besides, the parallelism of the watermarking algorithm is also vital for the actual time consumption.

### C.5 IMPACT ON THE QUALITY OF GENERATED TEXT

Codable watermarks contain more complex information and have a larger impact on the quality of text generated by LLMs than non-codable watermarks. Therefore, it is necessary to ensure that the impact of the watermark on the quality of LLM-generated text is within an acceptable range for the deployment of watermarking algorithms. In the current work, we adopt the text perplexity (PPL) as an automated metric to measure the quality of LLM-generated texts. In future work, we will include more metrics such as semantic similarity or human evaluation for comprehensive evaluation of text quality.

## D POTENTIAL APPLICATION SCENES OF CODABLE LLM WATERMARKING

In this section, we analyze some potential application scenarios of codable LLM watermarking, and examine how our method adjusts the proxy-LM to adapt to the varying demands for watermarking techniques in different application scenarios.

### D.1 CORPORATE INTELLECTUAL PROPERTY PROTECTION

For service providers based on LLMs, generating texts with watermarks containing information related to the model, service, request, or user can effectively ensure that the generated texts can be traced and identified, thereby preventing the model from being used unreasonably or without authorization. Service providers can flexibly choose and combine the information to be included in the watermark to better protect their intellectual property. In this scenario, as the owner of LLMs, the service provider can choose to use the large model itself as a proxy-LM to minimize the impact of adding watermarks on the quality of the text, or prepare a small model similar to the large one to accelerate inference through methods like distillation, quantization, or pruning.

### D.2 USER-LEVEL COPYRIGHT PROTECTION

The discussion on the copyright of text generated by LLMs is also an interesting topic, as users may believe that their intellectual input when writing prompts gives them (at least part of) the copyright of the generated text. In such a situation, users can choose to reach an agreement with the LLMs service provider on a customized watermark algorithm for the user self (through customizing the proxy-LM or hash function). In the encoding stage, the service provider will generate text containing a specific watermark according to the user's exclusive watermark encoding algorithm. When the user wants to prove that a piece of text comes from him- or herself, the user can request the service provider to use his or her exclusive decoding algorithm to confirm whether it contains a personal watermark. To make this more credible, an independent third-party organization similar to the patent office can take the responsibility to manage and certify these customized watermark algorithms.

### D.3 OPEN WATERMARKING PROTOCOL

For the public, it is desirable to have a very convenient way to identify whether a text comes from a model and which model it comes from. We propose an idea based on an open watermark protocol to reduce the identification problems caused by scattered and differentiated watermarking methods adopted by different service providers. First, the protocol selects an open-source language model (such as GPT-2) as a proxy-LM, and then determines a unified and scalable message coding system to establish a publicly available watermark encoding and decoding algorithm. Any service provider that joins this protocol can use the watermark encoding algorithm to inject watermarks into the texts generated by their private models by extending the message coding system. In this way, the public can efficiently identify all models that join the protocol using a single decoding algorithm. The technical support for this idea to work is that our Balance-Marking leverages both the LLMs and the proxy-LM during the encoding stage, while in the decoding stage, only the proxy-LM is needed to restore the watermark information. This makes it possible to have multiple closed-source LLMs encoding and a single public model decoding. If as many service providers as possible join this protocol, identifying the source of the text will become an increasingly easy task, which can effectively alleviate the impact of LLM-generated texts on human community order and security.

### D.4 Relay Watermarking among Models

Model-generated text that is actually applied is usually not generated in one step, but may go through multiple users or multiple models for processing, such as expansion, polishing, translation, paraphrasing, and so on. If we want to track the complete production process of a text, not just obtain the information of the last model processing, it is feasible to ensure that the watermark is incrementally written in a relay form among different models. As discussed in the previous subsection, this requires an open watermark protocol and a scalable information encoding system. This also requires adding a processing step to our method: first extract the watermark message from the text input to the model, mix it with the message of the new model itself, and then add the new message to the watermark of the newly generated text. In this way, we can track the complete life cycle of machine-generated text. Of course, as discussed in the previous subsection, this also requires as many LLM service providers as possible to join the open watermark protocol.

## E Approximation during calculation $P_w$

In calculating $P_w$ of Vanilla-Marking and Balance-Marking, we perform the following two approximations.

**Excluding Term** $\frac{1}{|\mathcal{M}|} \sum_{m' \in \mathcal{M}} \log P_w(v|m', \mathbf{t}_{:(l-1)})$ **from Encoding** When implementing Algorithm 1, there is a need to compute $\frac{1}{|\mathcal{M}|} \sum_{m' \in \mathcal{M}} \log P_w(v|m', \mathbf{t}_{:(l-1)})$, which can be a time-consuming task. However, due to the inherent randomness of $P_w$, we hypothesize that this mean value almost remains constant under the Law of Large Numbers. This hypothesis is corroborated by our empirical studies (see Appendix E.1) that indicate that this value remains close to -11 with a relatively insignificant standard deviation (less than 0.05) across $v$. Hence, we can exclude this calculation from Algorithm 1 during encoding for efficiency.

**Substituting** $\log P_w(v|m, \mathbf{t}_{:(l-1)})$ **with** $\log \hat{P}_w(v|m, \mathbf{t}_{:(l-1)})$ **during Decoding** Similarly, we **substitute** $\log P_w(v|m, \mathbf{t}_{:(l-1)})$ **with** $\log \hat{P}_w(v|m, \mathbf{t}_{:(l-1)})$ **in Eq. (12)**. The difference between the two expressions equals $\log \sum_v \hat{P}_w(v|m, \mathbf{t}_{:(l-1)})$, which we assume to be almost constant. Our empirical findings (see Appendix E.2) support this claim. In the Vanilla-Marking approach, the standard deviation is as low as 0.002, with an average value of 11.4. Despite Balance-Marking having a slightly higher deviation (0.23) and a mean of 11.4, it still allows for the exclusion of this calculation to help speed up the computation process.

### E.1 Excluding Term $\frac{1}{|\mathcal{M}|} \sum_{m' \in \mathcal{M}} \log P_w(v|m', \mathbf{t}_{:(l-1)})$ from encoding

The randomness present within the design of the message function $P_w$ allows us to interpret expressions of $\log P_w(v|m', \mathbf{t}_{:(l-1)})$s in $\frac{1}{|\mathcal{M}|} \sum_{m' \in \mathcal{M}} \log P_w(v|m', \mathbf{t}_{:(l-1)})$ as independent and identically distributed (i.i.d.) random variables. Consequently, via the law of large numbers, the term $\frac{1}{|\mathcal{M}|} \sum_{m' \in \mathcal{M}} \log P_w(v|m', \mathbf{t}_{:(l-1)})$ remains approximately constant.

As a practical test of our hypothesis, we randomly selected 100 sequences of $\mathbf{t}_{:(l-1)}$ from human-written texts (specifically, the news-like subset of the C4 dataset (Raffel et al., 2019)) as well as watermarked texts. Afterward, for each $\mathbf{t}_{:(l-1)}$, we randomly picked 100 tokens $v$ from the vocabulary to calculate the standard deviation of $\frac{1}{|\mathcal{M}|} \sum_{m' \in \mathcal{M}} \log P_w(v|m', \mathbf{t}_{:(l-1)})$ across $v$.[11] We report the standard deviation and mean across $v$ (averaged over 100 instances of $\mathbf{t}_{:(l-1)}$s) in Table 2. Given the small standard deviation across the vocabulary, it's appropriate to exclude it in Algorithm 1.

---

[11]Since the solution $t_l$ depends on the model logit and message logit of $v$ under a given $\mathbf{t}_{:(l-1)}$, it's enough to concentrate on the variations across $v$.

Table 2: $\frac{1}{|\mathcal{M}|}\sum\limits_{m' \in \mathcal{M}} \log P_w(v|m', \mathbf{t}_{:(l-1)})$ on human-written texts and watermarked texts. The numbers are the mean $\pm$ and the standard deviation across $v$.

|  | Human-Written Texts | Watermarked Texts |
|---|---|---|
| Vanilla-Marking | -10.95 $\pm$ 0.0005 | -10.95 $\pm$ 0.0005 |
| Balance-Marking | -10.93 $\pm$ 0.0461 | -10.93 $\pm$ 0.0457 |

### E.2 SUBSTITUTING $\log P_w(v|m, \mathbf{t}_{:(l-1)})$ WITH $\log \hat{P}_w(v|m, \mathbf{t}_{:(l-1)})$ DURING DECODING

Consider that

$$\log P_w(v|m, \mathbf{t}_{:(l-1)}) = \log \hat{P}_w(v|m, \mathbf{t}_{:(l-1)}) - \log \sum_v \hat{P}_w(v|m, \mathbf{t}_{:(l-1)}), \tag{17}$$

in a manner akin to Appendix E.1, we hypothesize that $\log \sum_v \hat{P}_w(v|m, \mathbf{t}_{:(l-1)})$ may remain nearly constant. If so, the magnitude of $P_w(v|m, \mathbf{t}_{:(l-1)})$ can be represented by the term $\hat{P}_w(v|m, \mathbf{t}_{:(l-1)})$.

As Appendix E.1, the term $\log \hat{P}_w(v|m, \mathbf{t}_{:(l-1)})$ can be interpreted as i.i.d. random variables, hence by invoking the law of large numbers, for any $\epsilon$, we establish:

$$\lim_{|\mathcal{V}| \to \infty} P\left( \left| \frac{\sum_v \hat{P}_w(v|m, \mathbf{t}_{:(l-1)})}{|\mathcal{V}|} - \mu \right| > \varepsilon \right) = 0, \tag{18}$$

with $\mathcal{V}$ representing the vocabulary and $\mu$ is the mean.

According to the mean value theorem, a $c$ exists between $\sum_v \hat{P}_w(v|m, \mathbf{t}_{:(l-1)})$ and $\mu|\mathcal{V}|$ such that

$$\frac{\log \sum_v \hat{P}_w(v|m, \mathbf{t}_{:(l-1)}) - \log(\mu|\mathcal{V}|)}{\sum_v \hat{P}_w(v|m, \mathbf{t}_{:(l-1)}) - \mu|\mathcal{V}|} = \frac{1}{c}. \tag{19}$$

Through empirical experimentation,[12] it is observed that $\log \sum_v \hat{P}_w(v|m, \mathbf{t}_{:(l-1)})$ exceeds 10.5. Given that the size of vocabulary $|\mathcal{V}|$ is 50257, it implies $c > e^{10.5} > \frac{1}{2}|\mathcal{V}|$, leading to

$$|\log \sum_v \hat{P}_w(v|m, \mathbf{t}_{:(l-1)}) - \log \mu|\mathcal{V}|| < 2|\sum_v \frac{\hat{P}_w(v|m, \mathbf{t}_{:(l-1)})}{|\mathcal{V}|}) - \mu|. \tag{20}$$

Merging Eq. (18) and Eq. (20) confirms that, with high certainty, , $\log \sum_v \hat{P}_w(v|m, \mathbf{t}_{:(l-1)})$ approximates a constant.

To validate our hypothesis, we randomly select 100 $\mathbf{t}_{:(l-1)}$ as above, and randomly pick 100 $m$ from $\mathcal{M}$. The standard deviation and mean of $\log \sum_v \hat{P}_w(v|m, \mathbf{t}_{:(l-1)})$ are presented in Table 3. The deviation of $\log \sum_v \hat{P}_w(v|m, \mathbf{t}_{:(l-1)})$ is higher in Balance-Marking than in Vanilla-Marking, due to the likelihood that $\log \hat{P}_w(v|m, \mathbf{t}_{:(l-1)})$ may not be precisely i.i.d for Balance-Marking. Nevertheless, this modest variation still allows us to exclude $\log \sum_v \hat{P}_w(v|m, \mathbf{t}_{:(l-1)})$ for the sake of swifter computation.

---

[12] We use the same experimental settings as the experiments shown in Table 3.

Table 3: $\log \sum_v \hat{P}_w(v|m, \mathbf{t}_{:(l-1)})$ on human-written texts and watermarked texts. The numbers are the average $\pm$ the standard deviation across $(\mathbf{t}_{:(l-1)}, m)$ .

|  | Human-Written Texts | Watermarked Texts |
|---|---|---|
| Vanilla-Marking | $11.45 \pm 0.0019$ | $11.45 \pm 0.0022$ |
| Balance-Marking | $11.42 \pm 0.1797$ | $11.41 \pm 0.2266$ |

## F    DETAILED ILLUSTRATIONS ABOUT THE PRACTICAL IMPROVEMENTS ON BALANCE-MARKING

The method we introduced at the beginning of Section 4.2.2 may encounter several obstacles in realistic applications. Thus, we make the following improvements to make it more practical when facing various realistic circumstances:

**(1) Omit $\mathbf{x}^{prompt}$ and truncate $\mathbf{t}_{:(l-1)}$ into a fixed length.**    Considering that the $\mathbf{x}^{prompt}$ is usually unavailable and the text receiver might only obtains a segment of the watermarked text during the message decoding phase, it will cause inconsistency between the encoding and decoding phases on calculating the token probabilities for creating $V_{m,\mathbf{t}_{:(l-1)}}$. To address this problem, we employ $P_{LLM}(v|\mathbf{t}_{(l-1-L_{prefix}):(l-1)})$ to approximate $P_{LLM}(v|\mathbf{x}^{prompt}, \mathbf{t}_{:(l-1)})$. That is, we omit $\mathbf{x}^{prompt}$ and truncate $\mathbf{t}_{:(l-1)}$ to a fixed-length $\mathbf{t}_{(l-1-L_{prefix}):(l-1)}$ for consistency during both encoding and decoding.

**(2) Use a proxy language model (proxy-LM) $LM_{proxy}$ in Eq. (9).**    Moreover, considering several usage scenarios discussed in Appendix D, we broaden the $LLM$ used in $P_{LLM}(v|\mathbf{t}_{(l-1-L_{prefix}):(l-1)})$ into a general proxy model denoted as $LM_{proxy}$. This model can either be $LLM$ itself when the model company wants the watermarked text only to be decoded by itself, or be another smaller and public language model (e.g., GPT-2 (Radford et al., 2019)) $P_{LLM}$ that allows for quicker computation of $P_w$ and enables anyone to decode the embedded message without knowing the specific $LLM$ used in text generation.

**(3) Pre-map message space into a smaller space for efficient computing.**    Since computing $V_{m,\mathbf{t}_{:(l-1)}}$ for each $m$ during encoding can be much time-consuming when the message space is pretty large, we opt to first map the entire message space into a smaller space as $m \to \hat{h}(m) \in \mathcal{M}_A = \{1, \cdots, A\}$ by using another hash function $\hat{h}$, and then compute the seed $s$ as $s = h(\hat{h}(m), x_{:(l-1)})$. By this way, we only need to run Algorithm 2 a mere $A$ times for each $\mathbf{t}_{:(l-1)}$.

## G    HYPERPARAMETERS AND HASH SCHEME

In Section 5.1, we opt to use $LM_{proxy} = $ GPT2 (Radford et al., 2019), $A = 100$, $L_{prefix} = 10$, $\sigma = 0.5$ and $\mathcal{M} = \{0, 1, ..., 2^{20} - 1\}$ for Balance-Marking. The reason to set $\sigma = 0.5$ is to achieve the maximal diversity of $V_{m,\mathbf{t}_{:(l-1)}}$ w.r.t. different $m$. The choice of $LM_{proxy}$, $A$, $L_{prefix}$, $\mathcal{M}$ aims to balance efficiency and performance, which is further discussed in Appendix G.1 to G.4. The hash scheme is detailed in Appendix G.5

### G.1    THE IMPACT OF THE PROXY LANGUAGE MODEL $LM_{proxy}$

In our Balance-Marking method, we employ $LM_{proxy}$ to generate an estimation for $P_{LLM}(v|\mathbf{t}_{:(l-1)})$. It can be reasonably hypothesized that an enhanced $LM_{proxy}$ could offer a more accurate estimation of $P_{LLM}(v|\mathbf{t}_{:(l-1)})$, thereby leading to superior performance. Experiments detailed in Figure 3 support this hypothesis.[13] However, GPT2-Large illustrates better performance in comparison to GPT2-XL, suggesting that the improvement might have a certain ceiling.

---

[13]Here, we run experiments with $\delta \in \{1.0, 1.2, 1.5, 2.0, 3.0\}$. Appendix G.2 to G.4 also use the same $\delta$s.

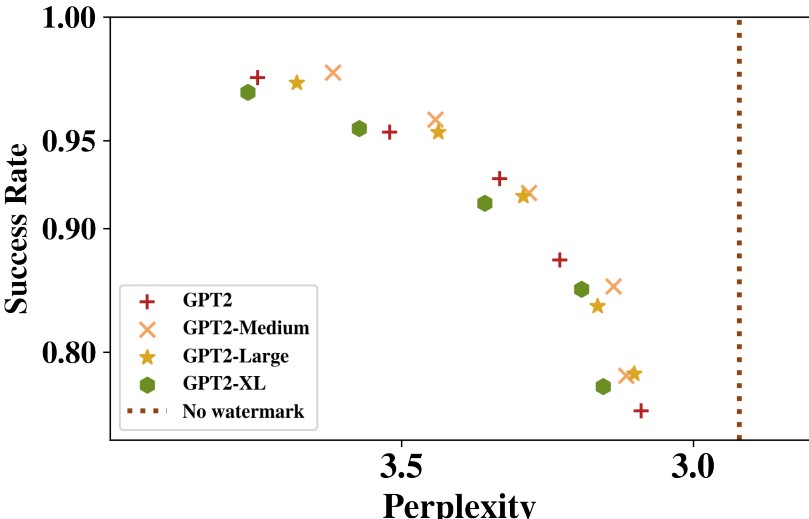

Figure 3: Illustration of $LM_{proxy}$'s impact on watermarking. Larger $LM_{proxy}$ tends to exhibit improved performance.

Table 4: Growing $LM_{proxy}$ size amplifies the computational cost during both encoding and decoding processes.

| $LM_{proxy}$ | GPT2 (124M) | GPT2-Medium (355M) | GPT2-Large (774M) | GPT2-XL (1.5B) |
|---|---|---|---|---|
| Encoding Time (s) | 9.50 | 11.69 | 14.08 | 16.38 |
| Decoding Time (s) | 2.97 | 3.09 | 3.43 | 3.53 |

Unfortunately, as the size of $LM_{proxy}$ increases, there follows a consequential surge in computational cost, clearly demonstrated in Table 4. For practical applications, it becomes necessary to establish a balance between time expenditure and the quality of the watermark.

**Special Case:** $LM_{proxy} = LLM$ Table 5 and Table 6 show the efficiency and quality of different $LM_{proxy}$s and the $LM_{proxy} = LLM$ case. $LM_{proxy} = LLM$ has the best quality, but is slower than $LM_{proxy} = $ GPT2.

### G.2 THE IMPACT OF THE PRE-MAPPING SPACE SIZE $A$

The purpose of introducing hyper-parameter $A$ in Balance-Marking is to reduce the number of $V_{m,\mathbf{t}_{:(l-1)}}$ to be computed. A lower value of $A$ results in lower computational expenses during decoding, as shown in Table 7. On the other hand, the cost of encoding remains relatively constant because only a single message $m$ requires the computation of $V_{m,\mathbf{t}_{:(l-1)}}$ during encoding.

To explore the impact of $A$ on watermarking quality, we conducted additional experiments as demonstrated in Figure 4. Although a larger $A$ generally leads to enhanced performance, there exist notable exceptions, such as when $A = 150$, where the performance is lower than when $A = 100$.

To strike a balance between performance and computational efficiency, we elected to use $A = 100$ for our main experiments in Section 5.2.

### G.3 THE IMPACT OF THE TRUNCATION LENGTH $L_{prefix}$

The parameter $L_{prefix}$, which is part of Eq. (11), impacts the quality of the watermark in two significant ways. (1) A longer $\mathbf{t}_{(l-1-L_{prefix}):(l-1)}$ in $P_{LM_{proxy}}(v|\mathbf{t}_{(l-1-L_{prefix}):(l-1)})$ can poten-

Table 5: Efficiency of different $LM_{proxy}$s and the $LM_{proxy} = LLM$ case.

| $LM_{proxy}$ | GPT2 (124M) | GPT2-Medium (355M) | GPT2-Large (774M) | GPT2-XL (1.5B) | LLM self (OPT-1.3B) |
|---|---|---|---|---|---|
| Encoding Time (s) | 9.50 | 11.69 | 14.08 | 16.38 | 11.05 |
| Decoding Time (s) | 2.97 | 3.09 | 3.43 | 3.53 | 4.07 |

Table 6: The watermark success rate of different $LM_{proxy}$s and the $LM_{proxy} = LLM$ case under different PPLs.

| PPL | 3.2 | 3.35 | 3.5 |
|---|---|---|---|
| GPT2 (124M) | 83.1 | 91.3 | 95.3 |
| GPT2-Medium (355M) | 84.9 | 93.2 | 95.1 |
| GPT2-Large (774M) | 88.6 | 93.8 | 96.7 |
| GPT2-XL (1.5B) | 86.4 | 93.4 | 96.0 |
| LLM self (OPT-1.3B) | 91.7 | 94.8 | 96.9 |

tially provide a superior approximation of $P_{LLM}(v|\mathbf{x}^{prompt}, \mathbf{t}_{:(l-1)})$. However, (2) an increased $\mathbf{t}_{(l-1-L_{prefix}):(l-1)}$ length will reduce the number of effective tokens in $\mathbf{t}$ available for encoding and decoding. This reduction occurs because $\mathbf{t}_{:L_{prefix}}$ lacks sufficient preceding words to form a $\mathbf{t}_{(l-1-L_{prefix}):(l-1)}$, which consequently results in their exclusion from encoding and decoding. Figure 5 illustrates the influence of $L_{prefix}$. Generally speaking, a relatively low $L_{prefix}$ value can degrade the watermark quality, while a moderate $L_{prefix}$ like 10 achieves similar performance to larger values such as 15 or 20.

The time cost for encoding and decoding remains relatively consistent across varying $L_{prefix}$ values, as demonstrated in Table 8. Interestingly, a larger $L_{prefix}$ might lead to a decrease in time cost, aligning with the aforementioned explanation that a more extended $L_{prefix}$ results in a shorter effective $\mathbf{t}$ utilized in the encoding and decoding stages.

### G.4 THE IMPACT OF THE MESSAGE SPACE SIZE $|\mathcal{M}|$

Increasing the size of $|\mathcal{M}|$ under a fixed payload information coding rate results in a larger number of tokens available for embedding a single message. In text watermarking scenarios, certain parts of the text often exhibit low entropy, thereby challenging the watermark encoding process (Kirchen. et al., 2023a). However, an increase in the number of tokens mitigates this problem, as it improves the likelihood of having high-entropy sections where the message can be effectively embedded. Moreover, let's consider an example where $|\mathcal{M}|$ is relatively small. In such a case, a piece of information of 20 bits might need to be divided into four separate chunks of 5 bits each to be encoded. If encoding fails for any of these 5-bit information segments, the encoding of the entire 20-bit information fails. This phenomenon potentially results in a significant accumulation of errors.

Figure 6 validates the analysis above by demonstrating a strong correlation between larger $|\mathcal{M}|$ and better watermark quality. Here, the experiment utilizes the $LLM$ to generate 210 tokens for encoding and decoding, since when $|\mathcal{M}| = 2^7$ and the coding rate is 10 tokens per bit, one message corresponds to 70 tokens.

Table 9 shows the time costs of the encoding and decoding processes when $|\mathcal{M}|$ increases. The time expenditure for encoding remains relatively stable, since there is only one message for which the calculation of $P_w$ is necessary. Contrastingly, the decoding process experiences a noticeable increase in time cost when $|\mathcal{M}|$ rises up to $2^{20}$. Thus, while an increase in $|\mathcal{M}|$ steadily improves watermark quality, the size of $\mathcal{M}$ can not be increased unlimitedly.[14]

---

[14]For instance, if $|\mathcal{M}|$ equals $2^{40}$, it becomes exceedingly difficult to calculate $P_w(\mathbf{t}|m')$ in Eq. (12) for all messages $m' \in \mathcal{M}$ during the decoding process.

Table 7: Investigation into the impact of $A$ on watermark efficiency. A larger $A$ value generally results in a longer decoding time.

| $A$ | 25 | 50 | 100 | 150 | 250 |
|---|---|---|---|---|---|
| Encoding Time (s) | 9.39 | 9.48 | 9.50 | 9.22 | 9.37 |
| Decoding Time (s) | 1.89 | 2.27 | 2.97 | 4.49 | 6.24 |

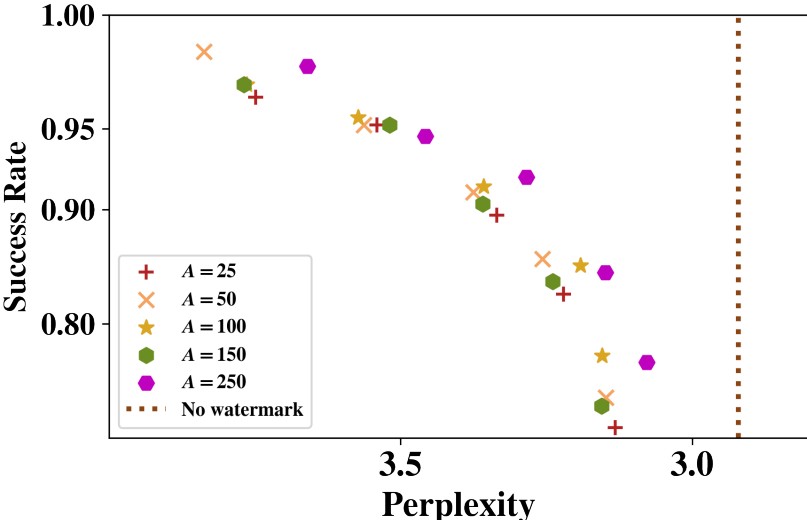

Figure 4: Illustration of $A$'s impact on watermarking quality. A higher $A$ tends to have better watermark quality.

### G.5  HASH SCHEME

Following the hash implementation in Kirchen. et al. (2023a), in the case of both the Vanilla-Marking and Balance-Marking methods, we make use of the last token in $\mathbf{t}_{:(l-1)}$ to calculate $h(v, m, \mathbf{t}_{:(l-1)})$ in Eq. (7) and $h(m, \mathbf{t}_{:(l-1)})$ in Algorithm 2, i.e. $h(v, m, \mathbf{t}_{:(l-1)}) = h(v, m, t_{l-1})$ and $h(m, \mathbf{t}_{:(l-1)}) = h(m, t_{l-1})$.

## H  IMPLEMENTATION OF SUBSTITUTION ATTACKS

For a chosen text to be tested on, we arbitrarily pick an unaltered token each time. This token is then masked and the model is asked to predict it. In the event that a token's predicted logit surpasses the predicted logit of the original token minus 1.0, we replace the original token with the new one. With a designated substitution ratio $\alpha$ and a sentence consisting of $L$ tokens, we continue this process until the substituted tokens reach the value of $\alpha L$ or after $3\alpha L$ attempts are made. From empirical analysis, we establish that such replacements result in a marginal increase in PPL by around 0.1.

## I  SUBSTITUTION ATTACKS WITH A SUBSTITUTION RATIO OF $10\%$

Figure 7 compares the results under a substitution ratio of 5% and 10%. A higher substitution results in a lower success rate, and the performance of Balance-Marking and Vanilla-Marking becomes closer from Figure 7a to Figure 7b, indicating that Substitution Attacks may do more hurt to Balance-Marking than Vanilla-Marking. This can be attributed to the fact that $P_w$ of Balance-Marking relies on $\mathbf{t}_{(l-1-L_{prefix}):(l-1)}$ (see Eq. (11)), while $P_w$ of Vanilla-Marking only depends on $t_{(l-1)}$, since we only use the last token of $\mathbf{t}_{:(l-1)}$ to calculate $h(m, \mathbf{t}_{:(l-1)})$ (see Section 5.1). So, Balance-Marking will be affected more when more tokens are substituted.

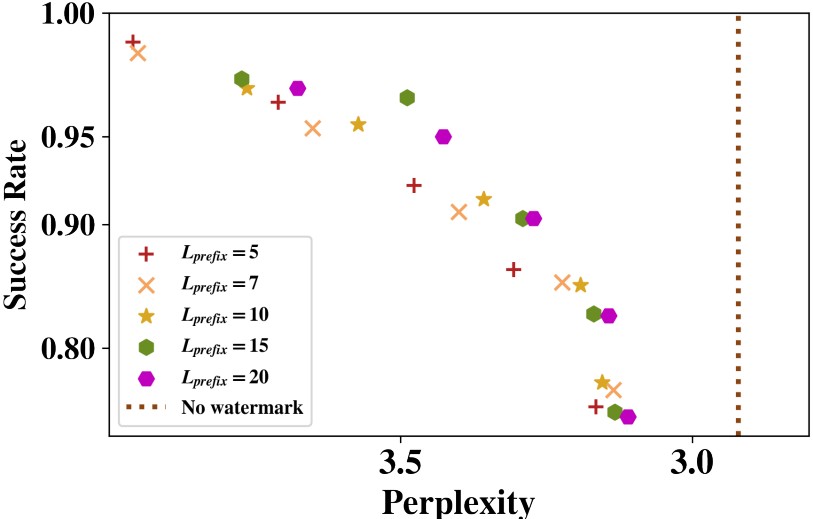

Figure 5: Illustration of $L_{prefix}$'s impact on watermark quality. A too-low $L_{prefix}$ value can degrade the watermark quality.

Table 8: Investigation into the impact of $L_{prefix}$ on watermark efficiency. $L_{prefix}$ has a modest impact on watermark efficiency.

| $L_{prefix}$ | 5 | 7 | 10 | 15 | 20 |
|---|---|---|---|---|---|
| Encoding Time (s) | 9.52 | 9.03 | 9.50 | 9.29 | 9.10 |
| Decoding Time (s) | 3.01 | 3.05 | 2.97 | 2.91 | 2.86 |

## J  RESULTS OF LLAMA-7B AND LLAMA-13B AT A CODING RATE OF 5

Besides experiments with a coding rate of 10 in Section 5.4, we also tested the coding rate of 5. The results are shown in Figure 8 and 9. Balance-Marking outperforms Vanilla-Marking in all cases.

## K  ILLUSTRATIVE EXAMPLES OF WATERMARKED TEXTS

In this section, we present some examples generated using our watermarking approach (Table 10). These examples serve as a case study to demonstrate the quality of our watermarking technique. All the generated sentences are truncated at 200 tokens.

Table 9: Investigation into the impact of $|\mathcal{M}|$ on watermark efficiency. The size of $\mathcal{M}$ has a modest impact on watermark efficiency.

| $|\mathcal{M}|$ | $2^5$ | $2^7$ | $2^{10}$ | $2^{20}$ |
|---|---|---|---|---|
| Encoding Time (s) | 9.89 | 10.02 | 9.83 | 10.05 |
| Decoding Time (s) | 2.37 | 2.34 | 2.36 | 3.13 |

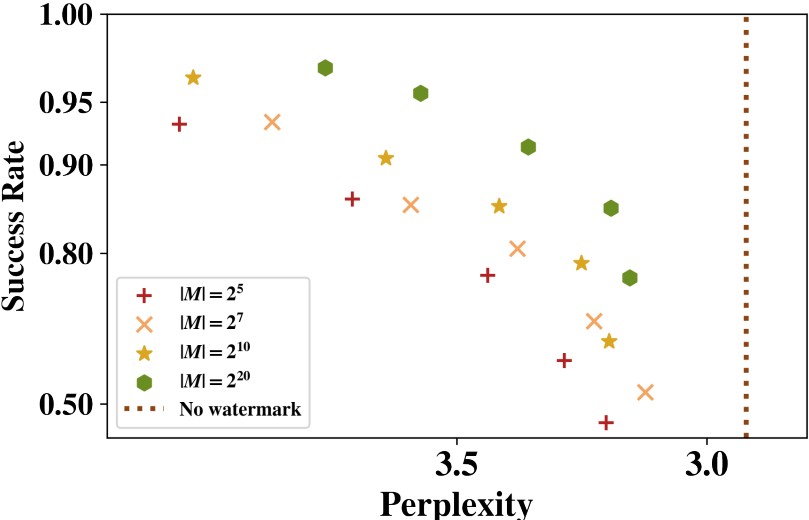

Figure 6: Illustration of $|\mathcal{M}|$'s impact on watermark quality. A larger $\mathcal{M}$ results in better watermark quality.

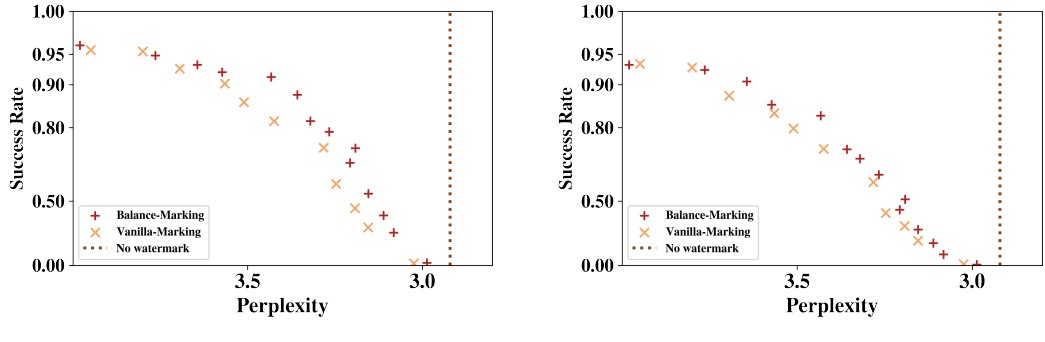

(a) Substitution Attacks (substitution ratio = 5%).     (b) Substitution Attacks (substitution ratio = 10%).

Figure 7: The relationship between the success rate after Copy-Paste Attacks / Substitution Attacks and the PPL. Balance-Marking outperforms Vanilla-Marking under both attacks.

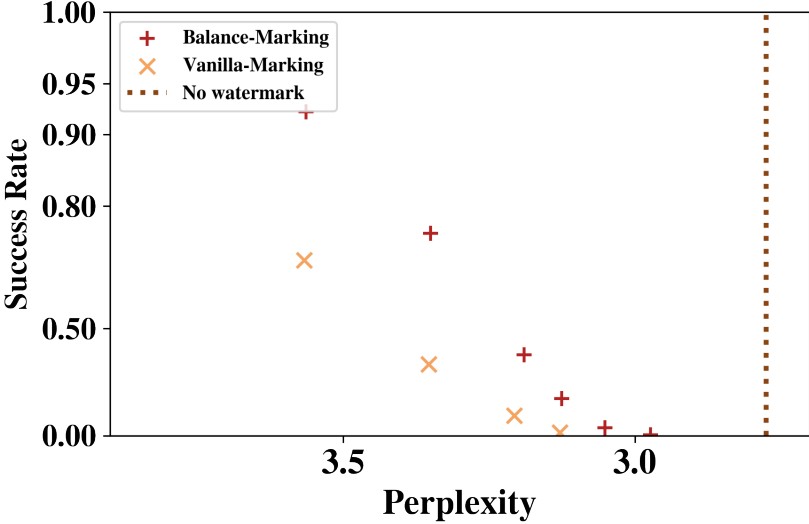

Figure 8: Balance-Marking outperforms Vanilla-Marking under LLaMA-7B and coding rate of 5 tokens per bit).

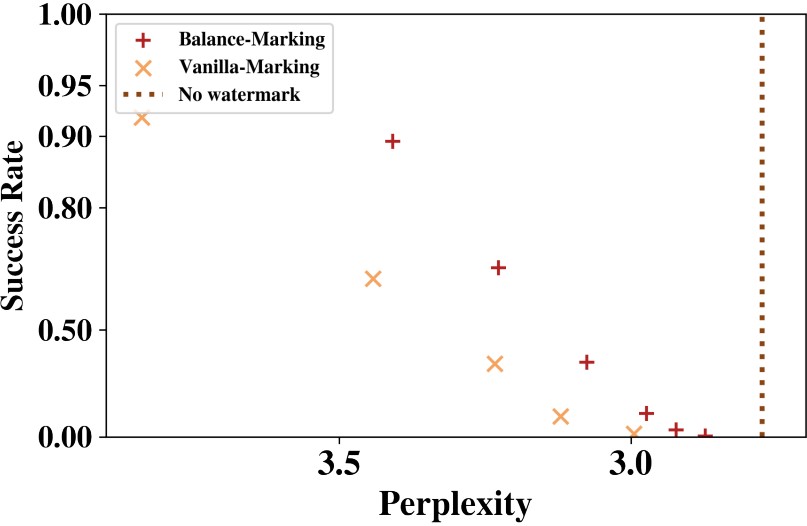

Figure 9: Balance-Marking outperforms Vanilla-Marking under LLaMA-13B and coding rate of 5 tokens per bit).

Table 10: Examples of generated texts.

| Case Study | Text |
|---|---|
| Prompt | An EgyptAir jet sent a series of signals indicating that smoke had been detected on board before it crashed into the Mediterranean on Thursday, France's air accident investigation agency said on Saturday. A spokesman for the BEA agency said the signals did not indicate what caused the smoke or fire on board the plane, which plunged into the sea with 66 people on board as it was heading from Paris to Cairo. But they offered the first clues as to what unfolded in the moments before the crash. One aviation source said that a fire on board would likely have generated multiple warning signals, while a sudden explosion may not have generated any. The army spokesman published pictures on Saturday on its official Facebook page of the recovered items, which included blue debris with EgyptAir markings, seat fabric with designs in the airline's colours, and a yellow lifejacket. It was the third blow since October to Egypt's travel industry, still reeling from political unrest following the 2011 uprising that ousted Hosni Mubarak. A suspected Islamic State bombing brought down a Russian airliner after it took off from Sharm el-Sheikh airport in late October, killing all 224 people on board, and an EgyptAir plane was hijacked in March by a man wearing a fake suicide belt. "These messages do not allow in any way to say what may have caused smoke or fire on board the aircraft," said a spokesman for the French BEA agency, which is assisting an official Egyptian |
| No-watermark (OPT-1.3B) | inquiry into the crash. "We will continue our work until we are able to give you more information," he added. Egyptian authorities have so far given few details about the crash, saying only that it happened shortly after take-off at 4:18 p.m. (1318 GMT) on Thursday. The flight data recorder and cockpit voice recorder were recovered from the ocean floor on Friday. The black box recorders are expected to be handed over to investigators later this month. BEA spokesman Jean-Francois Delfraissy told a news conference on Saturday that the black boxes contained "a great deal of information". He declined to comment on whether there were any indications of mechanical failure. He also said the BEA had received no reports of any distress calls from the plane before it disappeared from radar screens. French President Francois Hollande said on Saturday that his thoughts were with the families of those who died in the crash. |
| Vanilla-Marking (OPT-1.3B) (watermark success rate = 93%) | probe. "We will continue our work until we can be certain about the cause." Egyptian officials said earlier on Saturday that they had found no evidence of foul play in the crash. They also said there were no signs of mechanical failure on the Airbus A320, which was flying at 31,000 feet (9,800 meters) when it disappeared from radar screens shortly after take-off from Oran, near the Libyan border, at 2:24 p.m. local time (0824 GMT). The wreckage was found some 20 kilometres (12.5 miles) off the Libyan coast on Saturday afternoon. It is unclear how long it has been in the water or how deep it is. The search is being conducted by a naval patrol vessel, two ships from the Italian navy, one from the United Arab Emirates, one from Greece and a helicopter from the French Navy. The aircraft was carrying 149 passengers and nine crew members, including 82 Egyptians, 12 Italians, |

Table 10 (Continued): Examples of generated texts.

| Case Study | Text |
|---|---|
| Balance-Marking (OPT-1.3B) (watermark success rate = 95%) | inquiry into the crash. "They are just indications that there could have been some kind of activity on the part of the aircraft," he added. He declined to comment further. French President Francois Hollande, who visited the crash site in the southern city of Ismailia with Prime Minister Manuel Valls on Saturday, has called for an international investigation into the crash. Egypt's civil aviation authority said in a statement on Saturday that an initial examination of the black box flight data recorder showed that "there were no signs of a mechanical problem with the aircraft." It added: "The preliminary findings of the investigation suggest that there was a technical fault that led to the loss of communication between the aircraft and the ground control centre." CAUGHT BY SURVIVORS Survivors described hearing a loud noise and then seeing flames coming out of the rear of the Airbus A320, one of the world's most modern planes |
| No-watermark (LLaMA-7B) | aring a fake suicide belt who demanded to be flown to Cyprus. Egyptian President Abdel Fattah al-Sisi declared three days of national mourning and ordered flags to fly at half-mast until Tuesday. "We will spare no effort to find out the reasons behind this disaster," Prime Minister Sherif Ismail told reporters in Cairo. France's foreign minister Jean-Marc Ayrault said he had spoken to his Egyptian counterpart Sameh Shoukry and expressed France's "solidarity" with the families of the victims. French President Francois Hollande is due to speak to Sisi later on Saturday, according to the Elysee presidential palace. German Chancellor Angela Merkel said she was "deeply shaken" by the news of the crash, adding: "Our thoughts are with the families |
| Vanilla-Marking (LLaMA-7B) (watermark success rate = 96.8%) | aring a fake suicide belt who forced it to land in Cyprus. Egyptian officials say there is no evidence so far of a terrorist attack but investigators are treating the incident as a "criminal act". France's foreign minister Jean-Marc Ayrault, speaking on BFM TV, said he could not rule out a terrorist attack. "I can't exclude anything at this stage ... We must remain very vigilant," he said, adding that French experts were taking part in the investigation. Egyptian security officials told Reuters on Friday that a technical glitch was being investigated as the possible cause of the crash. Another senior Egyptian security official, speaking on condition of anonymity, said investigators had found no trace of explosives or traces of chemicals used to make explosive devices on the remains of the Airbus A320. Investigators also found |
| Balance-Marking (LLaMA-7B) (watermark success rate = 96.2%) | aring a fake suicide belt who demanded to be flown to Cyprus. He was overpowered and the plane landed safely in Cyprus. British Foreign Secretary Philip Hammond said he had spoken to Egyptian President Abdel Fattah al-Sisi about the incident. "I expressed our condolences to the families of the victims," he told reporters in Brussels. Hammond declined to comment when asked whether Britain would review security arrangements at British airports, including London's Heathrow, which has direct flights to Egypt's main tourist destinations. The United Nations Security Council condemned the "outrageous and cowardly terrorist attack" but made no mention of possible retaliatory action against militants in Sinai. The council also urged all states to cooperate with the Egyptian authorities in their efforts to bring perpetrators to justice. |

Table 10 (Continued): Examples of generated texts.

| Case Study | Text |
|---|---|
| No-watermark (LLaMA-13B) | aring a fake suicide belt. Egyptian Prime Minister Sherif Ismail told reporters at the crash site: "There is no indication so far of any terrorist or sabotage act." He added that investigators were looking into all possible causes including mechanical failure, human error and weather conditions. "We are working very hard to find the black box," he said, referring to the cockpit voice recorder and flight data recorder that could help explain why the Airbus A320 plummeted from 37,000 feet (11,280 metres) into the sea. French President Francois Hollande, whose country lost 54 citizens in the disaster, has ordered an inquiry to be carried out jointly with Egyptian authorities. The BEA will lead the French side of the probe. The BEA spokesman said the search for the wreckage |
| Vanilla-Marking (LLaMA-13B) (watermark success rate = 98%) | aring a suicide belt who forced it to divert to Cyprus. Egyptian Prime Minister Sherif Ismail told reporters on Saturday that investigations were focusing on the possibility of a "terrorist attack" but stressed this could not yet be confirmed. "There is no clear indication at this time," he said, adding that Egypt was cooperating with other countries to find out the cause of the crash. Investigators are combing through the wreckage of the Airbus (AIR.PA) A320 found 295 km north of the coastal city of Alexandria, searching for the black box flight recorders that will provide crucial clues to the cause of the crash. France's Bureau d'Enquetes et d'Analyses pour la Securite du Trafic Aérien (BEA), which is leading the probe into the crash, said |
| Balance-Marking (LLaMA-13B) (watermark success rate = 97%) | aring a fake suicide belt who forced it to divert to Cyprus. He surrendered and was arrested after giving himself up. The cause of Thursday's crash remains unknown, but the focus has turned to the possibility of a technical failure, terrorism or a deliberate act by the pilot or co-pilot, given their high level of training. The Airbus A320 is a workhorse of worldwide aviation. It has a good safety record, with only two fatal accidents in the past 15 years – one of them the Germanwings disaster in the French Alps last year, when a co-pilot appears to have intentionally crashed the plane, killing all 150 people on board. The other was an A320 operated by Indonesian budget carrier Adam Air that crashed into the sea off the coast of Sulawesi in 2007, |

