# OpenReview forum: "Towards Codable Watermarking for Injecting Multi-Bits Information to LLMs"
_ICLR.cc/2024/Conference — ICLR 2024 poster_

### Official Review · Reviewer_qLTC · 2023-10-25

**Soundness:** 3 good
**Presentation:** 4 excellent
**Contribution:** 2 fair
**Rating:** 6
**Confidence:** 3

**Summary:**

This work proposes a watermarking scheme for encoding multiple bits of information into LLM’s generated text. The scheme is built on a recent LLM watermarking work that encodes only one bit of information [Kirchenbauer, 2023a]. The main contribution of this work is showing a simple way to create balanced “green/red lists” using a public proxy LLM. The scheme performs better than the baseline used in Kirchenbauer [2023a] but suffers a large computation overhead.

---

### Update after rebuttal

I decided to raise my rating from 3 to 6 but decreased my confidence from 4 to 3. I'm also raising the soundness score from 2 to 3. My detailed response can be found at [this comment](https://openreview.net/forum?id=JYu5Flqm9D&noteId=xrgL53bRP8).

**Strengths:**

### Originality

The formulation leading up to Eq. (5) and (6) is neat in my opinion. It gives a nice interpretation of the “bias” in the logits as a Lagrangian multiplier from the constraint in Eq. (4). I have my doubts (mentioned later) about relying on perplexity as a metric in practice, but this does not take away the theoretical formulation here.

The idea of using a proxy LLM to help partition the green/red lists (”Balance-Marking”) is a nice idea and is the main contribution of this work. It seems to improve over the default random partition (”Vanilla-Marking”) empirically in all cases.

### Clarity

The main idea of the scheme and the experimental setup as well as the results are all conveyed clearly and effectively. I had no trouble (as far as I’m aware) following the paper.

**Weaknesses:**

Despite the aforementioned strengths, the paper has some weaknesses; I will try to list the ones that, I think, are more minor and easier to fix first.

### Perplexity as a proxy for text quality

It is well-known that repetitive texts can achieve very low perplexity, especially on older and weaker models [1,2]. I think there are better alternatives. First, I would prefer to see the perplexity computed by larger and better models than OPT-2.7B. I understand that Kirchenbauer [2023a] also uses perplexity as their main metric (also by OPT-2.7B) as well, but they take another alternative to show the text quality which is to simply include a good number of samples in the paper for the readers to judge. To be even more convincing, I like the approach taken by Fernandez et al. [2023] which is to use multiple text generation benchmarks that do not rely on perplexity. Other alternatives include using human evaluation or an oracle model (like GPT-3.5/4) for judging the similarity of watermarked vs non-watermarked texts.

### Theoretical bound on the watermarking efficacy

Kirchenbauer [2023a] provides a nice theorem (Theorem 4.2) that allows one to theoretically estimate the $p$-value of the watermark as a function of the spiked entropy (as well as other parameters). I believe that the same type of theorem can be derived for the proposed scheme in the multi-bit watermark as well. I believe that this component will significantly strengthen the paper.

### Metrics

It is mentioned that “success rate” combines both message recovery and watermark detection. First, I would like to confirm that a successful sample only counts if all 20 bits are correctly recovered and the watermark is correctly detected.

More importantly, I think there are two missing metrics: empirical $p$-value and the notion of FPR or AUC. Since watermarking is a sensitive application and a false positive can be extremely costly, these two metrics are particularly important. The paper briefly mentions a score threshold on $P_w(m|x)$, but it is simply fixed at 1e-5, and I could not find an explanation on how it is computed. This seems to give a “confidence score” on a particular message $m$ but does not look like an appropriate statistic for determining whether text $x$ is watermarked.

### Comparison to steganography

The paper briefly mentions steganography and some older watermarking works, but the experiments do not compare the proposed scheme to any. I believe that steganography and watermarking share a very similar concept; steganography is also used to convey multiple bits of information which makes it very relevant to this work. I would suggest that the authors try to compare the proposed scheme to steganography on natural language such as [3] and [4] as well as Yoo et al. [2023].

### Computation cost

Balance vocab partitioning comes with a great computation cost for decoding or verifying the watermark. The authors are aware of the limitation and have devised some mechanisms to reduce the overheads (using the second hash function and using a small proxy LLM). However, there are still two concerns:

1. Balance-marking with GPT-2 is twice as expensive as the baseline which I assume is already quite higher than the non-watermarked generation (it will be good to see this number too). This is a huge practical limitation.
2. More importantly, the fact that the linearly increased cost only yields diminishing improvement in the success rate is concerning (Figure 2a). The success rate seems still too low for practical use even with the heavy overheads.

**Minors/Typos**

- In Eq. (9), it seems like $v$ is missing. So it should be $P_{LLM}(v|...)$  instead of $P_{LLM}(...)$ .
- I believe that the variant of the copy-paste attack used in the experiment is fairly weak. A real attacker would interleave multiple parts of the non-watermarked text. This further reduces the effective number of watermarked tokens.
- Paraphrase attack is a common attack in the watermarking literature at this point. This attack is stronger than both the copy-paste and the synonym substitution used in the experiments. It is also easy to carry out in practice so I think it might be a good idea to include it.
- Section 5.5: in the first paragraph of this section, “$1 - 10^{-5}$” should be “$1 \times 10^{-5}$” instead.

**References**

- Kirchenbauer [2023a]: https://arxiv.org/abs/2301.10226
- Fernandez et al. [2023]: https://arxiv.org/abs/2308.00113
- Yoo et al. [2023]: https://aclanthology.org/2023.acl-long.117/
- [1] https://arxiv.org/abs/1904.09751
- [2] https://openreview.net/forum?id=SJeYe0NtvH
- [3] https://arxiv.org/abs/1909.01496
- [4] https://arxiv.org/abs/2210.14889

**Questions:**

- Page 4: When I read the last paragraph about using the *mean* of the log probability instead of the *max* overall messages $m' \in \mathcal{M}$, I was a little confused. More precisely, I didn’t see why it addresses the mentioned problem about “…solving $\hat m$ is infeasible because the true $\hat m$ can only be solved after the whole output t is determined…”; both options still rely on computing $\log P_w(t_l|m',\mathbf t_{:(t-1)})$ for all $m'$. What am I missing here?
- Is there any result on different hash lengths (i.e., how many of the previous tokens are used to compute hash)? This seems like an important experiment since it can affect both the robustness and the text quality (likelihood of repetition).
- I wonder how the proposed watermark performs if the proxy LLM is the LLM we want to watermark. Perhaps, the watermark’s efficiency can be improved significantly if this is the case, but it does limit a lot of use cases of the watermark as the authors have pointed out.
- I find it very interesting that larger models achieve a higher success rate (i.e., LLaMA-13B > LLaMA-7B > OPT-1.3B). This seems counterintuitive because larger and better models usually have lower perplexity/entropy so it should be more difficult to watermark them. I wonder if this is a pleasant side effect of the balance marking, but it seems like the success rate under the vanilla marking also improves. I wonder if the authors have any comment on this observation.
- What does “coding rate” exactly refer to? I assume that it means the number of watermarked tokens divided by the message length in bits, but it is mentioned that the text length is fixed to 200 and 20 bits for the message. So I could be misunderstanding this.

---

> ### Author Response · Authors · 2023-11-23
>
> **Q1:** Regarding the text quality metric:
>
>
>
> **A1:** We chose ppl since it is widely accepted for evaluating the quality of text generation.[1,2] Besides, in experiments for llama-7b/13b, we use a much larger model, llama-33b to calculate ppl, which may avoid some problems you mentioned about ppl. Also, we provide a case study to show the text quality [The case study is attached in another comment due to the character limitation of Openview].
>
>
>
> [1] https://arxiv.org/abs/2306.17439
>
>
>
> [2] https://arxiv.org/abs/2307.16230
>
>
>
> **Q2:** Regarding the theoretical bound.
>
>
>
> **A2:** Thank you for your suggestion. The derivation of a theoretical bound for the single-bit case is straightforward, as it involves simply counting the 'green tokens'. However, in multi-bit scenarios, the challenge lies in calculating the likelihood that any message, except for the accurate one, possesses a greater number of 'green tokens' than the accurate message. This calculation tends to be complex and often yields theoretical bounds that aren't practically applicable. We will try to explore this further in the future to see if any good bound can be derived.
>
>
>
> **Q3:** Regarding the questions about watermark accuracy metrics: **(1)** Does a successful sample only count if all 20 bits are correctly recovered and the watermark is correctly detected **(2)** The threshold of $1-10^{-5}$ and AUC metric.
>
>
>
> **A3:** To address your points:
>
>
>
> **(1)**. Yes, a sample is deemed successful only when all 20 bits are accurately recovered, and the watermark is correctly detected.
>
>
>
> **(2)**. Similar to the reason we discussed in Q2, p-value is not directly calculable in multi-bit watermarking scenarios as they are in single-bit ones. Instead, we compute $P_w(m|\mathbf{x}) $ to serve as a confidence score for whether a given message $m$ is watermarked in text $\mathbf{x}$. This is achieved through the following steps:
>
>
>
>   a. We first calculate $P_w(\mathbf x|m) = \sum_{l=1}^L \log P_w\left(x_l \mid m^{\prime}, \mathbf{x}_{:(l-1)}\right)$, the latter can be direclty computed based on our design of P_w.
>
>
>
>   b. By applying Bayes' Theorem, $P_w(m|\mathbf{x}) $ is proportional to $P_w(\mathbf{x}|m) $. With the normalization of $P_w(\mathbf{x}|m) $ over all $m $s, we derive $P_w(m|\mathbf{x}) $.
>
>
>
> For the threshold of $1- 1 \times 10^{-5} $: Our tests on 10,000 non-watermarked, human-written texts indicate that $\arg\max_m P_w(m|\mathbf{x}) $ falls below this threshold, suggesting it's stringent enough to avoid the false positives you mentioned (FPR=0). (We think it is enough since Kirchenbauer [2023a] verified an FPR=0 on a smaller sample of 500 human-written texts.) We extract the message only when it is above this threshold, ensuring no human-written text is falsely flagged as watermarked.
>
>
>
> Lastly, since multi-bit scenarios aren't binary classifications, AUC is not applicable. We focus on the success rate of correctly extracting messages when above the threshold, where no human-written text is misidentified as watermarked
>
>
>
> **Q4:** Regarding the comparison to steganography.
>
>
>
> **A4:** Steganography methods have to know the prompt before the generated text so as to get extract model logits when extracting message. This is unavailable in the watermark setting, making their transferring to watermark scenarios impossible. (To somewhat, this also indicates watermark is harder than steganography).
>
>
>
> For Yoo et al. [2023], this is a post-processing method. It does not integrate watermarks with LLM generation. It performs watermark using masked language models, rather than much powerful LLMs, inevitably leading to poor text quality. In experiments, we found this method tends to perform trivial and possibly inappropriate substitutions. Here are some cases (the original token is shown in the brackets after the changed token):
>
>
>
> The KPA **was(is)** the military arm of the ruling Workers' Party of Korea.
>
>
>
> The U.S. **military(government)** spends more money than it collects in tax revenues.
>
>
>
> "There is still a lot **less(of)** work to be done."
>
>
>
> You can also find such situations in the case study provided by Yoo et al. [2023] in their appendix.
>
>
>
> Yoo et al. [2023]: https://aclanthology.org/2023.acl-long.117/

---

> > ### Author Response · Authors · 2023-11-23
> >
> > **Q5:** Regarding the question about the computation cost of CTWL.
> >
> >
> >
> > **A5:** Embedding multi-bit watermarks is inherently more intrusive compared to single-bit watermarking, potentially degrading text quality more severely. There exists a trade-off and the increased cost is necessary if we want to achieve such a challenging task of encoding multi-bit information into the watermarked text without significant hurt to text quality, especially when we want no bit of the information is corrupted. Still, though balance-marking is slower than vanilla-marking, it achieves higher watermark success rate when we control text quality (ppl) to the same:
> >
> > | PPL (opt-2.7b)             | 3.3  | 3.4  | 3.5  | 3.6  |
> > | -------------------------- | ---- | ---- | ---- | ---- |
> > | Vanilla-Marking (opt-1.3b) | 81.2 | 85.9 | 90.5 | 93.6 |
> > | Balance-Marking (opt-1.3b) | 88.7 | 93.6 | 95.3 | 96.1 |
> >
> > | PPL (llama-33b)            | 3.2  | 3.3  | 3.4  | 3.5  |
> > | -------------------------- | ---- | ---- | ---- | ---- |
> > | Vanilla-Marking (llama-7b) | 73.1 | 87.9 | 93.2 | 96.2 |
> > | Balance-Marking (llama-7b) | 93.6 | 95.7 | 97.0 | 98.0 |
> >
> > | PPL (llama-33b)             | 3.0  | 3.1  | 3.2  | 3.4  |
> > | --------------------------- | ---- | ---- | ---- | ---- |
> > | Vanilla-Marking (llama-13b) | 45.4 | 75.4 | 88.1 | 96.4 |
> > | Balance-Marking (llama-13b) | 86.4 | 93.9 | 97.7 | 99.1 |
> >
> >
> >
> > The gap in watermark sucess rate between balance-marking and vanilla-marking indicating that in many cases when we focus on copyright protection or text traceability, the extra time cost is acceptable.
> >
> >
> >
> > **Q6:** Regarding the question that the success rate in Figure 2a seems still too low for practical use.
> >
> >
> >
> > **A6:** This success rate is under ppl of 3.4. Higher success rate can be achieved if we allow a little higher ppl. For the powerful llama-13b, it can achieve 99.2% when ppl evaluated by llama-33b is 3.42 (no watermark ppl is 2.78).
> >
> >
> >
> > **Q7:** Regarding the question "Why using the mean of the log probability instead of the max overall messages in the last paragraph of Page 4".
> >
> >
> >
> > **A7:** To address the reviewer's comment with precision, it appears there might have been a miscommunication in our manuscript. We'd like to clarify that the mean can indeed be computed straightforwardly, or in fact, regarded as a constant within the proper (and intuitively natural) design of $P_w$ . For instance, if we set $P_w$ to be completely random, as in Vanilla-Masking, then the mean $\frac{1}{|\mathcal{M}|} \sum_{m' \in \mathcal{M}} \log P_w(t_l | m', \mathbf{t}_{:(l-1)})$ assumes a constant value as per the law of large numbers. This concept is elaborated upon in Appendix E.1 of our paper. We hope this clarification resolves any confusion and accurately conveys the intended message.
> >
> >
> >
> > **Q8:** Regarding different hash lengths.
> >
> >
> >
> > **A8:** The selection of hash length is a compatible but separate consideration from our method, hence it was not a priority in our investigation. We have adopted the approach used by Kirchenbauer [2023a], opting for a hash token length of one for simplicity in this initial exploration. Additionally, we recognize that increasing hash length could lead to a higher susceptibility to substitution attacks. We intend to delve into more sophisticated hash schemes in subsequent research.
> >
> >
> >
> > **Q9:** Regarding the question "Will using $LM_{proxy}=LLM$ improve efficiency?"
> >
> >
> >
> > **A9:** Currently, our approach uses $ P_{LM_{proxy}}(v|\mathbf t_{(l-1-L_{prefix}):(l-1)}) $. If $LM_{proxy}$ is the same LLM, we still need to compute this alongside the normal $P_{LLM}(v\mid\mathbf t_{:(l-1)})$ in generation. Using $P_{LLM}(v\mid\mathbf t_{:(l-1)})$ instead of $P_{L M_{\text {proxy }}}(v \mid \mathbf t_{(l-1-L_{\text{prefix}}):(l-1)})$ may cause mismatch during watermark decoding, though it is promising in improving efficiency.
> >
> >
> >
> > **Q10:** Regarding the comment "Larger models achieve a higher success rate (i.e., LLaMA-13B > LLaMA-7B > OPT-1.3B) in this paper. This seems counterintuitive because larger and better models usually have lower perplexity/entropy so it should be more difficult to watermark them. "
> >
> >
> >
> > **A10:** There seems to be a misunderstanding regarding the interpretation of perplexity (ppl) scores. It is important to distinguish between comparisons of ppl within the same language model (LM) versus across different LMs. Within the same LM, a lower ppl might suggest a lack of diverse alternatives during text generation. However, comparing ppl scores between different LMs does not necessarily support the same conclusion. A more advanced LM could indeed yield a lower ppl for high-quality human-written text, signifying its proficiency. Yet, this does not mean it lacks the capability to offer a variety of alternative generations. In fact, a more sophisticated LM may be more adept at producing a range of plausible alternatives than its less advanced counterparts.

---

> > > ### Author Response · Authors · 2023-11-23
> > >
> > > **Q11:** Regarding the question "What does “coding rate” exactly refer to? I assume that it means the number of watermarked tokens divided by the message length in bits, but it is mentioned that the text length is fixed to 200 and 20 bits for the message."
> > >
> > >
> > >
> > > **A11:** Yes, the term "coding rate" in our paper refers to the number of watermarked tokens divided by the message length in bits. Indeed, for the sake of clarity and consistency across our experiments, we have fixed the text length at 200 tokens and the message length at 20 bits. This standardization allows for a straightforward comparison of results.
> > >
> > >
> > >
> > > Still, these parameters can be adjusted depending on specific requirements. For example, we also included results where the coding rate is 5 tokens per bit, as shown in Figure 1(b) and Appendix J, Figure 8. In such scenarios, the model is tasked with generating 200 tokens, with the intent to embed a 20-bit message within the first and the last 100 tokens, respectively. For watermark extraction, we retrieve 20 bits from each 100-token section and concatenate them, resulting in a complete 40-bit message.
> > >
> > >
> > >
> > > **Q12:** Regarding the experiments on stronger attacks.
> > >
> > >
> > >
> > > **A12:** In this paper, we demonstrated that our method is robust to two commonly used attacks: copy-paste attacks and substitution attacks. However, it is a common and great challenge for all current watermark methods to defend more stronger attacks like paraphrase attacks. [1] Ensuring the robustness of multi-bit watermarks against paraphrasing attacks or the multi-copy-paste attack you mentioned is an even more significant challenge than the one-bit situation, since we can not allow one bit of the message to be corrupted. Admittedly, both vanilla-marking and balance-marking can not defend against such attacks well, and this issue is designated for exploration in our future research.
> > >
> > >
> > >
> > > [1] https://arxiv.org/pdf/2303.11156.pdf
> > >
> > >
> > >
> > > **Q13:** Regarding the comment "Section 5.5: in the first paragraph of this section, “1−10−5” should be “1×10−5” instead."
> > >
> > >
> > >
> > > **A13:** It appears there might be some confusion regarding this threshold. In A3, we provided a detailed explanation of the threshold value. I would recommend referring back to A3 for this.
> > >
> > >
> > >
> > > **Q14:** For typos in Eq. 9.
> > >
> > >
> > >
> > > **A14:** Thank you very much, we will revise it in the revision.

---

> > > > ### Author Response · Authors · 2023-11-23
> > > >
> > > > |         Case Study                                                 | Text                                                         |
> > > > | -------------------------------------------------------- | ------------------------------------------------------------ |
> > > > | Prompt                                                   | An EgyptAir jet sent a series of signals indicating that smoke had been detected on board before it crashed into the Mediterranean on Thursday, France’s air accident investigation agency said on Saturday. A spokesman for the BEA agency said the signals did not indicate what caused the smoke or fire on board the plane, which plunged into the sea with 66 people on board as it was heading from Paris to Cairo. But they offered the first clues as to what unfolded in the moments before the crash. One aviation source said that a fire on board would likely have generated multiple warning signals, while a sudden explosion may not have generated any. The army spokesman published pictures on Saturday on its official Facebook page of the recovered items, which included blue debris with EgyptAir markings, seat fabric with designs in the airline’s colours, and a yellow lifejacket. It was the third blow since October to Egypt’s travel industry, still reeling from political unrest following the 2011 uprising that ousted Hosni Mubarak. A suspected Islamic State bombing brought down a Russian airliner after it took off from Sharm el-Sheikh airport in late October, killing all 224 people on board, and an EgyptAir plane was hijacked in March by a man wearing a fake suicide belt. “These messages do not allow in any way to say what may have caused smoke or fire on board the aircraft,” said a spokesman for the French BEA agency, which is assisting an official Egyptian |
> > > > | No-watermark(opt-1.3b)                                   | inquiry into the crash. “We will continue our work until we are able to give you more information,” he added. Egyptian authorities have so far given few details about the crash, saying only that it happened shortly after take-off at 4:18 p.m. (1318 GMT) on Thursday. The flight data recorder and cockpit voice recorder were recovered from the ocean floor on Friday. The black box recorders are expected to be handed over to investigators later this month. BEA spokesman Jean-Francois Delfraissy told a news conference on Saturday that the black boxes contained “a great deal of information”. He declined to comment on whether there were any indications of mechanical failure. He also said the BEA had received no reports of any distress calls from the plane before it disappeared from radar screens. French President Francois Hollande said on Saturday that his thoughts were with the families of those who died in the crash. |
> > > > | Vanilla-Marking(opt-1.3b) (watermark success rate = 93%) | probe. “We will continue our work until we can be certain about the cause.” Egyptian officials said earlier on Saturday that they had found no evidence of foul play in the crash. They also said there were no signs of mechanical failure on the Airbus A320, which was flying at 31,000 feet (9,800 meters) when it disappeared from radar screens shortly after take-off from Oran, near the Libyan border, at 2:24 p.m. local time (0824 GMT). The wreckage was found some 20 kilometres (12.5 miles) off the Libyan coast on Saturday afternoon. It is unclear how long it has been in the water or how deep it is. The search is being conducted by a naval patrol vessel, two ships from the Italian navy, one from the United Arab Emirates, one from Greece and a helicopter from the French Navy. The aircraft was carrying 149 passengers and nine crew members, including 82 Egyptians, 12 Italians, |
> > > > | Balance-Marking(opt-1.3b) (watermark success rate = 95%) | inquiry into the crash. “They are just indications that there could have been some kind of activity on the part of the aircraft,” he added. He declined to comment further. French President Francois Hollande, who visited the crash site in the southern city of Ismailia with Prime Minister Manuel Valls on Saturday, has called for an international investigation into the crash. Egypt’s civil aviation authority said in a statement on Saturday that an initial examination of the black box flight data recorder showed that “there were no signs of a mechanical problem with the aircraft.” It added: “The preliminary findings of the investigation suggest that there was a technical fault that led to the loss of communication between the aircraft and the ground control centre.” CAUGHT BY SURVIVORS Survivors described hearing a loud noise and then seeing flames coming out of the rear of the Airbus A320, one of the world’s most modern planes |

---

> > > > > ### Author Response · Authors · 2023-11-23
> > > > >
> > > > > | Case Study                                                  | Text                                                         |
> > > > > | ----------------------------------------------------------- | ------------------------------------------------------------ |
> > > > > | Prompt                                                      | An EgyptAir jet sent a series of signals indicating that smoke had been detected on board before it crashed into the Mediterranean on Thursday, France’s air accident investigation agency said on Saturday. A spokesman for the BEA agency said the signals did not indicate what caused the smoke or fire on board the plane, which plunged into the sea with 66 people on board as it was heading from Paris to Cairo. But they offered the first clues as to what unfolded in the moments before the crash. One aviation source said that a fire on board would likely have generated multiple warning signals, while a sudden explosion may not have generated any. The army spokesman published pictures on Saturday on its official Facebook page of the recovered items, which included blue debris with EgyptAir markings, seat fabric with designs in the airline’s colours, and a yellow lifejacket. It was the third blow since October to Egypt’s travel industry, still reeling from political unrest following the 2011 uprising that ousted Hosni Mubarak. A suspected Islamic State bombing brought down a Russian airliner after it took off from Sharm el-Sheikh airport in late October, killing all 224 people on board, and an EgyptAir plane was hijacked in March by a man we |
> > > > > | No-watermark(llama-7b)                                      | aring a fake suicide belt who demanded to be flown to Cyprus. Egyptian President Abdel Fattah al-Sisi declared three days of national mourning and ordered flags to fly at half-mast until Tuesday. “We will spare no effort to find out the reasons behind this disaster,” Prime Minister Sherif Ismail told reporters in Cairo. France’s foreign minister Jean-Marc Ayrault said he had spoken to his Egyptian counterpart Sameh Shoukry and expressed France’s “solidarity” with the families of the victims. French President Francois Hollande is due to speak to Sisi later on Saturday, according to the Elysee presidential palace. German Chancellor Angela Merkel said she was “deeply shaken” by the news of the crash, adding: “Our thoughts are with the families |
> > > > > | Vanilla-Marking(llama-7b)  (watermark success rate = 96.8%) | aring a fake suicide belt who forced it to land in Cyprus. Egyptian officials say there is no evidence so far of a terrorist attack but investigators are treating the incident as a “criminal act”. France’s foreign minister Jean-Marc Ayrault, speaking on BFM TV, said he could not rule out a terrorist attack. “I can’t exclude anything at this stage … We must remain very vigilant,” he said, adding that French experts were taking part in the investigation. Egyptian security officials told Reuters on Friday that a technical glitch was being investigated as the possible cause of the crash. Another senior Egyptian security official, speaking on condition of anonymity, said investigators had found no trace of explosives or traces of chemicals used to make explosive devices on the remains of the Airbus A320. Investigators also found |
> > > > > | Balance-Marking(llama-7b)  (watermark success rate = 96.2%) | aring a fake suicide belt who demanded to be flown to Cyprus. He was overpowered and the plane landed safely in Cyprus. British Foreign Secretary Philip Hammond said he had spoken to Egyptian President Abdel Fattah al-Sisi about the incident. “I expressed our condolences to the families of the victims,” he told reporters in Brussels. Hammond declined to comment when asked whether Britain would review security arrangements at British airports, including London’s Heathrow, which has direct flights to Egypt’s main tourist destinations. The United Nations Security Council condemned the “outrageous and cowardly terrorist attack” but made no mention of possible retaliatory action against militants in Sinai. The council also urged all states to cooperate with the Egyptian authorities in their efforts to bring perpetrators to justice. |

---

> ### Author Response · Authors · 2023-11-23
>
> | Case Study                                                | Text                                                         |
> | --------------------------------------------------------- | ------------------------------------------------------------ |
> | Prompt                                                    | An EgyptAir jet sent a series of signals indicating that smoke had been detected on board before it crashed into the Mediterranean on Thursday, France’s air accident investigation agency said on Saturday. A spokesman for the BEA agency said the signals did not indicate what caused the smoke or fire on board the plane, which plunged into the sea with 66 people on board as it was heading from Paris to Cairo. But they offered the first clues as to what unfolded in the moments before the crash. One aviation source said that a fire on board would likely have generated multiple warning signals, while a sudden explosion may not have generated any. The army spokesman published pictures on Saturday on its official Facebook page of the recovered items, which included blue debris with EgyptAir markings, seat fabric with designs in the airline’s colours, and a yellow lifejacket. It was the third blow since October to Egypt’s travel industry, still reeling from political unrest following the 2011 uprising that ousted Hosni Mubarak. A suspected Islamic State bombing brought down a Russian airliner after it took off from Sharm el-Sheikh airport in late October, killing all 224 people on board, and an EgyptAir plane was hijacked in March by a man we |
> | No-watermark(llama-13b)                                   | aring a fake suicide belt. Egyptian Prime Minister Sherif Ismail told reporters at the crash site: “There is no indication so far of any terrorist or sabotage act.” He added that investigators were looking into all possible causes including mechanical failure, human error and weather conditions. “We are working very hard to find the black box,” he said, referring to the cockpit voice recorder and flight data recorder that could help explain why the Airbus A320 plummeted from 37,000 feet (11,280 metres) into the sea. French President Francois Hollande, whose country lost 54 citizens in the disaster, has ordered an inquiry to be carried out jointly with Egyptian authorities. The BEA will lead the French side of the probe. The BEA spokesman said the search for the wreckage |
> | Vanilla-Marking(llama-13b) (watermark success rate = 97%) | aring a suicide belt who forced it to divert to Cyprus. Egyptian Prime Minister Sherif Ismail told reporters on Saturday that investigations were focusing on the possibility of a “terrorist attack” but stressed this could not yet be confirmed. “There is no clear indication at this time,” he said, adding that Egypt was cooperating with other countries to find out the cause of the crash. Investigators are combing through the wreckage of the Airbus (AIR.PA) A320 found 295 km north of the coastal city of Alexandria, searching for the black box flight recorders that will provide crucial clues to the cause of the crash. France’s Bureau d’Enquetes et d’Analyses pour la Securite du Trafic Aérien (BEA), which is leading the probe into the crash, said |
> | Balance-Marking(llama-13b) (watermark success rate = 98%) | aring a fake suicide belt who forced it to divert to Cyprus. He surrendered and was arrested after giving himself up. The cause of Thursday’s crash remains unknown, but the focus has turned to the possibility of a technical failure, terrorism or a deliberate act by the pilot or co-pilot, given their high level of training. The Airbus A320 is a workhorse of worldwide aviation. It has a good safety record, with only two fatal accidents in the past 15 years – one of them the Germanwings disaster in the French Alps last year, when a co-pilot appears to have intentionally crashed the plane, killing all 150 people on board. The other was an A320 operated by Indonesian budget carrier Adam Air that crashed into the sea off the coast of Sulawesi in 2007, |

---

> ### Comment · Reviewer_qLTC · 2023-12-01
>
> I apologize for the delayed response. I appreciate the author(s)'s efforts in putting together the rebuttal. All my questions were addressed with clarity, and I am very happy with the answers. I decided to raise my rating to 6; I believe that this is still a borderline case, and the work has two main limitations preventing it from being practical: (1) the watermarking efficiency/accuracy, and (2) the decoding cost and the requirement of a proxy LLM. Regardless, this is one of the first papers to tackle this problem (along with a couple of other concurrent works) so I am willing to overlook these limitations.
>
> Finally, I just want to add a few future suggestions in case the author(s) find them helpful. I'm not asking the author(s) to do these experiments at this point, but just want to throw out some ideas I got:
>
> 1. I can see a future improvement or a follow-up work trying to use a cheaper way to do the balancing instead of a proxy LLM. Vanilla and LLM-balancing are two ends of the spectrum in terms of the "balancing accuracy" as well as computation costs. I suspect that there is a way to strike a good trade-off here, i.e., with a tiny cheap model for balancing that may be "good enough" for watermarking. Exploring this trade-off may be very interesting.
> 2. I would love to see the number of message bits (or coding rate) that can be achieved with X% (say 5%) decrease in perplexity and Y%  (say 99.9%) success rate. I am more concerned about success rate (or accuracy) than perplexity because this type of task has a really high cost for an incorrect prediction whereas higher perplexity does not always mean a lower quality for some downstream tasks.

---

### Official Review · Reviewer_CxJT · 2023-10-31

**Soundness:** 3 good
**Presentation:** 3 good
**Contribution:** 3 good
**Rating:** 6
**Confidence:** 4

**Summary:**

This paper first theoretically analyses two main limitations of the Codable Text Watermarking for LLMs (CTWL) field: (1) encode limited information (only 1 bit) ;(2) ignore the quality of generated watermarking texts. Then, authors propose a advanced CTWL method named Balance-Marking. The core idea of our method is to use a proxy language model to split the vocabulary into probability-balanced parts, thereby effectively maintaining the quality of the watermarked text. Extensive experimental results show that our method outperforms the baseline under comprehensive evaluation.

**Strengths:**

1. This paper gives a general mathematical presentation for the watermarked text for LLM.
2. The proposed method comprehensively consider the trade-off between watermarked text quality and embedded capacity.

**Weaknesses:**

1. The details of the formulation presentation have some mistakes.
2. The robustness experiments are not enough.

**Questions:**

(1)	From Eq. (16) in Appendix, I cannot get the same formulation as Eq. (5). As we know, the Lagrange multipliers are commonly added to the constraint item.
(2)	Authors use Proxy-LM to approximate the condition defined by Eq. (9), which inevitably decreases the watermarking text quality. The main performance difference between Proxy-LM and the original LLM in Eq. (5) should be comprehensively discussed, including text quality, success rate, robustness, etc.
(3)	Authors just analyze the trade-off between efficiency and watermark success rate in different proxy-LMs, which makes me question that authors subjectively miss the trade-off between text quality and coding rate of payload information in different proxy-LMs. Moreover, the caption of Figure (2)(a) is inconsistent with the corresponding descriptions.
(4)	Authors do not evaluate the robustness of the proposed method to machine paraphrasing attacks which have been considered in some previous works (“Kirchen. et al., 2023a” and “Kirchen. et al., 2023b”).
(5)	It is analyzed that what are the different experimental results when using different sampling strategies? such as greedy sampling, top-p sampling, etc.

---

> ### Author Response · Authors · 2023-11-23
>
> **Q1:** Regarding how to obtain Eq. (5) from Eq. (16) in the Appendix.
>
>
>
> **A1:** The mathematical formulation is correct. In Eq. (16), we first let $\delta=\frac{L}{\lambda}$, and we can get
>
> $$ \max_{t} [ \sum_{l=1}^{L} \log P_{w}(t_{l}|m,t_{:(l-1)}) - \max_{m' \neq m} \sum_{l=1}^{L} \log P_{w}(t_{l}|m',t_{:(l-1)}) + \frac{1}{\delta} \sum_{l=1}^{L} \log P_{LLM}(t_{l}|x^{prompt},t_{:(l-1)}) ].$$ Then, let $\hat m=\arg\max_{m' \neq m}\sum_{l=1}^{L} \log P(t_{l}|m',t_{:(l-1)})$, the above target can be further formulated as $$ \max_{t} [ \sum_{l=1}^{L} \log P_{w}(t_{l}|m,t_{:(l-1)}) -  \sum_{l=1}^{L} \log P_{w}(t_{l}|\hat{m},t_{:(l-1)}) + \frac{1}{\delta} \sum_{l=1}^{L} \log P_{LLM}(t_{l}|x^{prompt},t_{:(l-1)}) ]. $$ This is equivalent to $$ \max_{t} [ \delta \sum_{l=1}^{L} (\log P_{w}(t_{l}|m,t_{:(l-1)}) -  \log P_{w}(t_{l}|\hat{m},t_{:(l-1)}) )+ \sum_{l=1}^{L} \log P_{LLM}(t_{l}|x^{prompt},t_{:(l-1)}) ], $$ which is just the same as Eq. (5).
>
>
>
> **Q2:** Regarding the performance difference between using a smaller proxy-LM in Eq. (11) and using the original LM in Eq. (9).
>
>
>
> **A2:** Thank you for this question, and we agree that there is a trade-off between the success rate, text quality and encoding efficiency when choosing proxy-LM. We included the original LM in the comparison of Figure 2(a), and derived the following table:
>
>
>
> | PPL         | 3.2  | 3.35 | 3.5  |
> | ----------- | ---- | ---- | ---- |
> | GPT2        | 83.1 | 91.3 | 95.3 |
> | GPT2-medium | 84.9 | 93.2 | 95.1 |
> | GPT2-large  | 88.6 | 93.8 | 96.7 |
> | GPT2-XL     | 86.4 | 93.4 | 96.0 |
> | orginal LM  | 91.7 | 94.8 | 96.9 |
>
>
>
> |             | Time for generation text | Time for extracting information |
> | ----------- | ------------------------ | ------------------------------- |
> | GPT2        | 9.5                      | 2.97                            |
> | GPT2-medium | 11.69                    | 3.09                            |
> | GPT2-large  | 14.08                    | 3.43                            |
> | GPT2-XL     | 16.38                    | 3.53                            |
> | orginal LM  | 11.05                    | 4.07                            |
>
>
>
>
>
> The original LM choice does perform better than smaller proxy-LMs, but is slower than small proxy-LMs like GPT2.
>
>
>
>
>
> **Q3:** Regarding the confusion about Figure 2(a): whether text quality is missing.
>
>
>
> **A3:** Apologies for any misunderstanding. To clarify, in Figure 2(a), we control the ppl to the same (3.4 here) and report the corresponding watermark success rate for different $LM_{proxy}$s. This approach ensures that the comparison takes into account text quality. We prioritize illustrating the variation in watermark success rates at a fixed text quality, which we believe may be of greater interest than the inverse relationship. Additionally, comprehensive results concerning text quality and watermark success rates can be found in Appendix G.1, illustrated in Figure 3. Regarding the caption for Figure 2(a), the term 'quality' was meant to encapsulate the balance achieved between text quality and watermark success rate. We acknowledge this may have caused confusion and will amend the caption to better convey this meaning.
>
> Besides, since we control the ppl and watermark success rate by $\delta$, achieving an exact ppl of 3.4 is challenging. To estimate the watermark success rate under a ppl of 3.4, we have employed linear interpolation between the two nearest (ppl, watermark success rate) points.
>
>
>
> **Q4:** Regarding the machine paraphrasing attacks.
>
>
>
> **A4:** In this paper, we demonstrated that our method is robust to two commonly used attacks: copy-paste attacks and substitution attacks. However, it is a common and great challenge for all current watermark methods to defend against paraphrase attacks. [1] Our method also suffers from this problem when we try to attack it with gpt3.5. Ensuring the robustness of multi-bit watermarks against paraphrasing attacks is an even more significant challenge than the one-bit situation, since we can not allow one bit of the message to be corrupted. This issue is designated for exploration in our future research.
>
>
>
> [1] https://arxiv.org/pdf/2303.11156.pdf
>
>
>
> **Q5:** Regarding the experiments with more sampling strategies like greedy sampling or top-p sampling:
>
>
>
> **A5:** While we acknowledge the potential benefits of exploring a broader range of sampling strategies, such as greedy or top-p sampling, constraints in time and resources have limited our capacity to conduct extensive experimentation in this area. We aim to expand our investigation into these strategies in the future version.

---

### Official Review · Reviewer_iBFF · 2023-11-01

**Soundness:** 2 fair
**Presentation:** 2 fair
**Contribution:** 2 fair
**Rating:** 6
**Confidence:** 2

**Summary:**

This work fill in the blank of watermark injection on LLM generated text on multi-bits information encoding during LLM generation. Prior work use Vanilla-marking to encode multi-bit information, yet they decrease generation quality. This work proposes a balance marking algorithm, the goal is to improve the generation quality, which can be achieved by making the LLM lose close to the original one. This work guides the vocabulary partitioning with a proxy-LM. Results are demonstrated on OPT, LLaMA-7B, 13B.

**Strengths:**

1. Interesting work on multi-bit injection to LLM while considering the text quality.

**Weaknesses:**

1. Writing needs improving. There is a lot of math and equation, and less intuition.

2. I do not understand the paper after a few passes, while I am not in the watermarking field, the content should be written so that it can be understood by the general ML audience.

**Questions:**

I am new to the field, and I I am still confused after reading twice. Can the author explain this in plain language?

The goal is to find the prompt that separate watermarked input and non-watermarked input the most right? Not separating machine-generated vs human-generated. As I see Eq 3 is separating both machine-generated text. There is no human written one.

How does the method encode multibit information? Is the information to be encoded is m? Eq 12 decodes M, yet this formulation is very similar to adversarial attack, why does the method can decode the unique m instead of some adversarial string? Like Zou et al. Universal and Transferable Adversarial Attacks on Aligned Language Models.

Does Model logit used for guard generation quality?

Does message logit used to add watermark?

Does a larger L give better watermark and higher quality?

I understand you want a v which create a large Pw differece for detection, does eq 7,9 simply find the words in a predefined dictionary?

What does sigma mean? In plain language, I do not intuitively understand.

How does Eq 10 encode the quality, Eq 9 inside?

I don't get why you need Eq 11.

---

> ### Author Response · Authors · 2023-11-23
>
> **Q1:** Regarding the writing and the intuition parts.
>
>
>
> **A1:** Thank you for your kind suggestion. We will proofread the paper carefully and try to add more intuition parts to make our paper more readable. In the following, we will answer your questions one by one to help you to better understand our paper.
>
>
>
> **Q2:** Regarding the question of whether the goal is to separate the watermarked text from unwatermarked text.
>
>
>
> **A2:** Yes, and to be more concrete, the goal of Eq. (3) is to separate the watermarked texts encoded with the message $m$ from other texts that are not encoded with $m$. However, our method can also distinguish watermarked machine-generated texts from human-written texts according to the results in Figure 2(b).
>
>
>
> **Q3**: Regarding the questions about the message encoding process.
>
>
>
> **A3:** Yes, the message we want to encode is $m$. Furthermore, each $m$ can be represented by a unique multi-bit 0-1 string from an entire message space $M= ${$0,...,2^{20}-1 $}. Then, during generating each token, we can use the above string of $m$ to calculate the random seed, calculate the message logit of each token, and split the vocabulary in a probability-balanced way. Finally, the next token is sampled only from the available part of the two splits. Therefore, as we can see, the generation process of each token depends on the information of $m$, which helps us to encode $m$ into the whole text.
>
>
>
> **Q4:** Regarding the question about the decoding process. Comparison with Zou et al. Universal and Transferable Adversarial Attacks on Aligned Language Models.
>
>
>
> **A4:** In the decoding process, the candidate string is only from the pre-defined set $M=${$0,...,2^{20}-1 $} . Therefore, the decoded string can only be one of the 0/1 strings in M rather than another arbitrary string like the adversarial string in Zou et.al. And, after obtaining the decoded 0/1 string, we can map it back to its original corresponding text message. We will cite and compare this paper in our future version.
>
>
>
> **Q5:** Regarding the question "Is the model logit used to guarantee the generation quality".
>
>
>
> **A5:** Yes, the model logit measures how likely is a specific token to be the next token, and higher model logits generally lead to higher text quality.
>
>
>
> **Q6:** Regarding the question "Is the message logit used to add watermark"?
>
>
>
> **A6:** Yes. The message logits are decided by the message $m$, and by adding the message to the model logits in the next token's generation procedure, we successfully encode the information of $m$ into the generated text.
>
>
>
> **Q7:** Regarding the question "Does a larger L give a better watermark and higher quality"？
>
>
>
> **A7:** Yes, refer to the answers in A5 and A6.
>
>
>
> **Q8:** Regarding the question "Do Eq. (7) and Eq. (9) simply find the words in a pre-defined vocabulary"?
>
>
>
> **A8:** Yes. This pre-defined vocabulary $V_{m,t_{:l-1}}$ contains the words that have high model logits, which ensures that we can sample a $v$ that not only has significantly higher message logits $P_{w}$ than that of other $v'$, but also is a reasonable word that is high likely to be the next token.
>
>
>
> **Q9:** Regarding the question "What does sigma mean".
>
>
>
> **A9:** $\sigma$ controls the probability of the tokens with high model logits to be included in the subset $V_{m,t_{:l-1}}$. Setting larger $\sigma$ will more likely include more tokens with high model logits in the available subset, but will also decrease the diversity of the splittings of $V_{m,t_{:l-1}}$ across different $m$s and make the message logit lose its original function.
>
>
>
> **Q10:** Regarding the question "How does Eq. (10) encode quality".
>
>
>
> **A10:** In Eq. (10), we only assign the model logits to the tokens that are included in the subset $V_{m,t_{:l-1}}$. As stated above and explained in the paragraph below Eq. (9) in the main paper, this subset should contain some tokens with relatively high model logits. Therefore, we ensure that the next token to maximize $L_{m,x^{prompt},t_{:l-1}}$ can be sampled from these reasonable tokens instead of arbitrary tokens caused by the random vocabulary splitting of Vanilla-Marking.

---

> > ### Author Response · Authors · 2023-11-23
> >
> > **Q11**: Regarding the question about Eq. (11).
> >
> >
> >
> > **A11:** Eq. (11) is derived from Eq. (9) by making two improvements to make Balanced-Marking more practical to deal with multiple application scenarios. (1) Firstly, Considering that the $x^{prompt}$ is usually unavailable to the text receiver during the message decoding phase, we omit $x^{prompt}$ and truncate $t_{:(l−1)}$ to a fixed-length $t_{(l−1−L_{prefix}):(l−1)}$ to make the encoding and decoding independent on the $x^{prompt}$. (2) Secondly, in order to apply Balance-Marking in various usage scenarios discussed in Appendix D, such as when the text receiver wants to recover the message from the text but does not have access to the internal logits of the original LLM, we broaden the original LLM used in $P_{LLM(v|t_{(l−1−L_{prefix}):(l−1)})}$ into a general proxy model denoted as LM_proxy. Therefore, designing Eq. (11) makes our Balanced-Marking more practical and effective. We have included a detailed discussion about the above points in Appendix F.

---

### Official Review · Reviewer_pLxG · 2023-11-02

**Soundness:** 2 fair
**Presentation:** 3 good
**Contribution:** 2 fair
**Rating:** 5
**Confidence:** 2

**Summary:**

In this paper, the authors propose a novel approach to inject multi-bit message information into large language models (LLMs). To inject the watermark, their approach enlarges the gap between the probability that the text is generated under the specific message and other messages. They also propose the balance-marking algorithm to maintain the text quality while injecting the watermark. Experiment results show that CTWL outperforms other baseline approaches in different dimensions.

**Strengths:**

1. It is the first work to inject the multi-bit information during the generation process of LLM instead of the postprocessing. The authors propose a novel approach, CTWL, to effectively encode the information while maintaining the text quality. CTWL proposes a Balance-Marking algorithm to consider that some generated tokens should have high model logits and message logits at the same time. Therefore, CTWL can inject the watermark into the generated text with only a slight reduction in the text quality.
2. Experiment results show that CTWL is robust against the copy-paste attacks and substitution attacks.
3. CTWL can also work for larger LLM (LLaMA-7/13B).

**Weaknesses:**

1. It is essential to compare the watermarked text with other approaches to evaluate the effectiveness of the proposed method. In the last paragraph of the related work section, the authors stated, “There are some very concurrent works ... the vocabulary partitions....” Still, there are not any experiments that compare the text quality between CTWL and other approaches. The authors are suggested to compare the BLEU, ROUGE, or other metrics with the previous studies, such as Kirchenbauer et al. (2023a), to show the text quality of CTWL is better than previous work. These experiments can make the results more convincing.

2. It would be better if the author could show some examples of watermarked texts to compare Balance-Marking and Vanilla-Marking. The case study can help the readers know which cases Balance-Marking works but Vanilla-Marking fails. The authors can also compare the encoding time of CTWL with previous studies to show that the encoding time of CTWL is reasonable and practical.

**Questions:**

1. How about the text quality of CTWL compared to the previous work, which focuses on injecting information by postprocessing?
2. Can the authors show some cases to explain how Balance-Marking works but Vanilla-Marking fails?
3. What is the computation cost of previous studies compared to CTWL?

---

> ### Author Response · Authors · 2023-11-23
>
> **Q1:** Regarding comparing CTWL with concurrent studies.
>
>
>
> **A1:** Actually, they were updating their papers a few days ago before the submission deadline, so we do not have time to compare them at that that time. And after studing the concurrent studies by Yoo et al.[1] and Fernandez et al.[2], we found that Vanilla-Marking can cover these works, since
>
>
>
> (1) Yoo et al. concentrates on bit-wise accuracy and strategies for bit allocation to text tokens, while our work is geared towards the accuracy of encoding and decoding entire messages. This is inherently more challenging as it is akin to raising bit-wise accuracy to the power of the number of bits in a message, which would result in a significantly lower performance under our metric for the study by Yoo et al. Thus, the bit allocation strategy of Yoo et al. can not be transferred to our settings, and their method, if excluded this allocation strategy, can be seen as equivalent to Vanilla-Marking.
>
>
>
> (2) Fernandez et al.'s research, on the other hand, aligns closely with what we refer to as Vanilla-Marking in our paper. Although they employ a circular-shift method to expedite the generation of a secret key, we have similarly optimized the hash function in our own implementations of Vanilla-Marking and Balance-Marking, as can be seen in our 'hash_fn.py' code. When considering text quality and watermark success rate, Fernandez et al.'s method should be same as Vanilla-Marking. Additionally, their scope is limited to messages in the range of 1-1000, while our approach extends to a much larger range of 1-2^20, (i.e., 1-1,048,576).
>
>
>
> [1]https://arxiv.org/abs/2308.00221
>
>
>
> [2]https://arxiv.org/abs/2308.00113
>
>
>
> **Q2:** Regarding using other metrics to compare the text quality between CTWL and other methods.
>
>
>
> **A2:** We adhere to the approach established by Kirchenbauer et al. (2023a) in utilizing perplexity (ppl) as a measure of text quality. Perplexity is widely accepted for evaluating the quality of text generation.[1,2] Alternatives like BLEU and ROUGE are less appropriate for text watermarking. The watermarking process inherently involves substituting texts to their alternatives, which may not align well with the n-gram overlap metrics that BLEU and ROUGE emphasize. Also, in experiments for llama-7b/13b, we use a much larger model, llama-33b to calculate ppl.
>
>
>
> [1] https://arxiv.org/abs/2306.17439
>
>
>
> [2] https://arxiv.org/abs/2307.16230
>
>
>
> **Q3:** Regarding the comparison between CTWL and the previous methods that inject information into text by post-processing.
>
>
>
> **A3:** We focus on watermark integrated with LLM generation, since the post-processing based methods will predictably lead to a low-quality machine-generated text. The major reason for this is that the abilities of masked-language-modeling-based models used in these post-processing-based methods (e.g., BERT) are far away from the current generative language models (e.g., LLaMa) for generating the texts, using incompetent models to replace the words in the sentences generated by powerful LLMs inevitably destroy the quality of the texts.
>
>
>
> We tried Yoo et al. [2023], a sota post-processing method that has also been mentioned by Reviewer qLTC. However, we find this method tends to perform trivial and possibly inappropriate substitutions, as we have discussed above. Here are some cases (the original token is shown in the brackets after the changed token):
>
>
>
> The KPA **was(is)** the military arm of the ruling Workers' Party of Korea.
>
>
>
> The U.S. **military(government)** spends more money than it collects in tax revenues.
>
>
>
> "There is still a lot **less(of)** work to be done."
>
>
>
> **Q4:** Regarding the comparison between the computation time of CTWL and that of previous methods.
>
>
>
> **A4:** Previous methods are not able to conduct multi-bit watermark and we are the first work to integrate it with LLM generation. So, there is no proper baseline that can be directly compared with. As an alternative choice, we view Vanilla-Marking as a baseline (though it is also proposed by us). We have included a comparison of the computational costs of Balance-Marking and Vanilla-Marking (proxy-LM = null) in Figure 2(a). (As we discussed in A1, the very concurrent work can be considered as an equivalent version of Vanilla-Marking, thus, Figure 2(a) also represents the comparison between Balance-Marking and concurrent methods.)
>
>
>
> It is worth noting that embedding multi-bit watermarks is inherently more intrusive compared to single-bit watermarking, potentially degrading text quality more severely. So, it is necessary for a trade-off between text quality and efficiency. Balance-Marking allows us to select different proxy-LMs to align varying preferences for quality and efficiency across diverse scenarios.

---

> > ### Author Response · Authors · 2023-11-23
> >
> > **Q5:** Regarding the cases studies about balance-marking and vanilla-marking.
> >
> >
> >
> > **A5:** We provide a case study in the rebuttal (The case study is attached in another comment due to character limitation of Openview). In the case study provided, we ensure that the watermarking success rates for both Vanilla-Marking and Balance-Marking are comparable. We then evaluate the texts they generate. Due to the prompt truncation at 300 tokens by the LLM's tokenizer, the resultant prompts for opt and llama differ in length. Both Vanilla-Marking and Balance-Marking produce text with good fluency when llama-7b/13b are used, while Balance-Marking tends to maintain better topical consistency with the original prompt. [The case study is attached in another comment due to character limitation of Openview]

---

> > > ### Author Response · Authors · 2023-11-23
> > >
> > > |         Case Study                                                 | Text                                                         |
> > > | -------------------------------------------------------- | ------------------------------------------------------------ |
> > > | Prompt                                                   | An EgyptAir jet sent a series of signals indicating that smoke had been detected on board before it crashed into the Mediterranean on Thursday, France’s air accident investigation agency said on Saturday. A spokesman for the BEA agency said the signals did not indicate what caused the smoke or fire on board the plane, which plunged into the sea with 66 people on board as it was heading from Paris to Cairo. But they offered the first clues as to what unfolded in the moments before the crash. One aviation source said that a fire on board would likely have generated multiple warning signals, while a sudden explosion may not have generated any. The army spokesman published pictures on Saturday on its official Facebook page of the recovered items, which included blue debris with EgyptAir markings, seat fabric with designs in the airline’s colours, and a yellow lifejacket. It was the third blow since October to Egypt’s travel industry, still reeling from political unrest following the 2011 uprising that ousted Hosni Mubarak. A suspected Islamic State bombing brought down a Russian airliner after it took off from Sharm el-Sheikh airport in late October, killing all 224 people on board, and an EgyptAir plane was hijacked in March by a man wearing a fake suicide belt. “These messages do not allow in any way to say what may have caused smoke or fire on board the aircraft,” said a spokesman for the French BEA agency, which is assisting an official Egyptian |
> > > | No-watermark(opt-1.3b)                                   | inquiry into the crash. “We will continue our work until we are able to give you more information,” he added. Egyptian authorities have so far given few details about the crash, saying only that it happened shortly after take-off at 4:18 p.m. (1318 GMT) on Thursday. The flight data recorder and cockpit voice recorder were recovered from the ocean floor on Friday. The black box recorders are expected to be handed over to investigators later this month. BEA spokesman Jean-Francois Delfraissy told a news conference on Saturday that the black boxes contained “a great deal of information”. He declined to comment on whether there were any indications of mechanical failure. He also said the BEA had received no reports of any distress calls from the plane before it disappeared from radar screens. French President Francois Hollande said on Saturday that his thoughts were with the families of those who died in the crash. |
> > > | Vanilla-Marking(opt-1.3b) (watermark success rate = 93%) | probe. “We will continue our work until we can be certain about the cause.” Egyptian officials said earlier on Saturday that they had found no evidence of foul play in the crash. They also said there were no signs of mechanical failure on the Airbus A320, which was flying at 31,000 feet (9,800 meters) when it disappeared from radar screens shortly after take-off from Oran, near the Libyan border, at 2:24 p.m. local time (0824 GMT). The wreckage was found some 20 kilometres (12.5 miles) off the Libyan coast on Saturday afternoon. It is unclear how long it has been in the water or how deep it is. The search is being conducted by a naval patrol vessel, two ships from the Italian navy, one from the United Arab Emirates, one from Greece and a helicopter from the French Navy. The aircraft was carrying 149 passengers and nine crew members, including 82 Egyptians, 12 Italians, |
> > > | Balance-Marking(opt-1.3b) (watermark success rate = 95%) | inquiry into the crash. “They are just indications that there could have been some kind of activity on the part of the aircraft,” he added. He declined to comment further. French President Francois Hollande, who visited the crash site in the southern city of Ismailia with Prime Minister Manuel Valls on Saturday, has called for an international investigation into the crash. Egypt’s civil aviation authority said in a statement on Saturday that an initial examination of the black box flight data recorder showed that “there were no signs of a mechanical problem with the aircraft.” It added: “The preliminary findings of the investigation suggest that there was a technical fault that led to the loss of communication between the aircraft and the ground control centre.” CAUGHT BY SURVIVORS Survivors described hearing a loud noise and then seeing flames coming out of the rear of the Airbus A320, one of the world’s most modern planes |

---

> > > > ### Author Response · Authors · 2023-11-23
> > > >
> > > > | Case Study                                                  | Text                                                         |
> > > > | ----------------------------------------------------------- | ------------------------------------------------------------ |
> > > > | Prompt                                                      | An EgyptAir jet sent a series of signals indicating that smoke had been detected on board before it crashed into the Mediterranean on Thursday, France’s air accident investigation agency said on Saturday. A spokesman for the BEA agency said the signals did not indicate what caused the smoke or fire on board the plane, which plunged into the sea with 66 people on board as it was heading from Paris to Cairo. But they offered the first clues as to what unfolded in the moments before the crash. One aviation source said that a fire on board would likely have generated multiple warning signals, while a sudden explosion may not have generated any. The army spokesman published pictures on Saturday on its official Facebook page of the recovered items, which included blue debris with EgyptAir markings, seat fabric with designs in the airline’s colours, and a yellow lifejacket. It was the third blow since October to Egypt’s travel industry, still reeling from political unrest following the 2011 uprising that ousted Hosni Mubarak. A suspected Islamic State bombing brought down a Russian airliner after it took off from Sharm el-Sheikh airport in late October, killing all 224 people on board, and an EgyptAir plane was hijacked in March by a man we |
> > > > | No-watermark(llama-7b)                                      | aring a fake suicide belt who demanded to be flown to Cyprus. Egyptian President Abdel Fattah al-Sisi declared three days of national mourning and ordered flags to fly at half-mast until Tuesday. “We will spare no effort to find out the reasons behind this disaster,” Prime Minister Sherif Ismail told reporters in Cairo. France’s foreign minister Jean-Marc Ayrault said he had spoken to his Egyptian counterpart Sameh Shoukry and expressed France’s “solidarity” with the families of the victims. French President Francois Hollande is due to speak to Sisi later on Saturday, according to the Elysee presidential palace. German Chancellor Angela Merkel said she was “deeply shaken” by the news of the crash, adding: “Our thoughts are with the families |
> > > > | Vanilla-Marking(llama-7b)  (watermark success rate = 96.8%) | aring a fake suicide belt who forced it to land in Cyprus. Egyptian officials say there is no evidence so far of a terrorist attack but investigators are treating the incident as a “criminal act”. France’s foreign minister Jean-Marc Ayrault, speaking on BFM TV, said he could not rule out a terrorist attack. “I can’t exclude anything at this stage … We must remain very vigilant,” he said, adding that French experts were taking part in the investigation. Egyptian security officials told Reuters on Friday that a technical glitch was being investigated as the possible cause of the crash. Another senior Egyptian security official, speaking on condition of anonymity, said investigators had found no trace of explosives or traces of chemicals used to make explosive devices on the remains of the Airbus A320. Investigators also found |
> > > > | Balance-Marking(llama-7b)  (watermark success rate = 96.2%) | aring a fake suicide belt who demanded to be flown to Cyprus. He was overpowered and the plane landed safely in Cyprus. British Foreign Secretary Philip Hammond said he had spoken to Egyptian President Abdel Fattah al-Sisi about the incident. “I expressed our condolences to the families of the victims,” he told reporters in Brussels. Hammond declined to comment when asked whether Britain would review security arrangements at British airports, including London’s Heathrow, which has direct flights to Egypt’s main tourist destinations. The United Nations Security Council condemned the “outrageous and cowardly terrorist attack” but made no mention of possible retaliatory action against militants in Sinai. The council also urged all states to cooperate with the Egyptian authorities in their efforts to bring perpetrators to justice. |

---

> > > > > ### Author Response · Authors · 2023-11-23
> > > > >
> > > > > | Case Study                                                | Text                                                         |
> > > > > | --------------------------------------------------------- | ------------------------------------------------------------ |
> > > > > | Prompt                                                    | An EgyptAir jet sent a series of signals indicating that smoke had been detected on board before it crashed into the Mediterranean on Thursday, France’s air accident investigation agency said on Saturday. A spokesman for the BEA agency said the signals did not indicate what caused the smoke or fire on board the plane, which plunged into the sea with 66 people on board as it was heading from Paris to Cairo. But they offered the first clues as to what unfolded in the moments before the crash. One aviation source said that a fire on board would likely have generated multiple warning signals, while a sudden explosion may not have generated any. The army spokesman published pictures on Saturday on its official Facebook page of the recovered items, which included blue debris with EgyptAir markings, seat fabric with designs in the airline’s colours, and a yellow lifejacket. It was the third blow since October to Egypt’s travel industry, still reeling from political unrest following the 2011 uprising that ousted Hosni Mubarak. A suspected Islamic State bombing brought down a Russian airliner after it took off from Sharm el-Sheikh airport in late October, killing all 224 people on board, and an EgyptAir plane was hijacked in March by a man we |
> > > > > | No-watermark(llama-13b)                                   | aring a fake suicide belt. Egyptian Prime Minister Sherif Ismail told reporters at the crash site: “There is no indication so far of any terrorist or sabotage act.” He added that investigators were looking into all possible causes including mechanical failure, human error and weather conditions. “We are working very hard to find the black box,” he said, referring to the cockpit voice recorder and flight data recorder that could help explain why the Airbus A320 plummeted from 37,000 feet (11,280 metres) into the sea. French President Francois Hollande, whose country lost 54 citizens in the disaster, has ordered an inquiry to be carried out jointly with Egyptian authorities. The BEA will lead the French side of the probe. The BEA spokesman said the search for the wreckage |
> > > > > | Vanilla-Marking(llama-13b) (watermark success rate = 98%) | aring a suicide belt who forced it to divert to Cyprus. Egyptian Prime Minister Sherif Ismail told reporters on Saturday that investigations were focusing on the possibility of a “terrorist attack” but stressed this could not yet be confirmed. “There is no clear indication at this time,” he said, adding that Egypt was cooperating with other countries to find out the cause of the crash. Investigators are combing through the wreckage of the Airbus (AIR.PA) A320 found 295 km north of the coastal city of Alexandria, searching for the black box flight recorders that will provide crucial clues to the cause of the crash. France’s Bureau d’Enquetes et d’Analyses pour la Securite du Trafic Aérien (BEA), which is leading the probe into the crash, said |
> > > > > | Balance-Marking(llama-13b) (watermark success rate = 97%) | aring a fake suicide belt who forced it to divert to Cyprus. He surrendered and was arrested after giving himself up. The cause of Thursday’s crash remains unknown, but the focus has turned to the possibility of a technical failure, terrorism or a deliberate act by the pilot or co-pilot, given their high level of training. The Airbus A320 is a workhorse of worldwide aviation. It has a good safety record, with only two fatal accidents in the past 15 years – one of them the Germanwings disaster in the French Alps last year, when a co-pilot appears to have intentionally crashed the plane, killing all 150 people on board. The other was an A320 operated by Indonesian budget carrier Adam Air that crashed into the sea off the coast of Sulawesi in 2007, |

---

### Meta-Review · Area_Chair_MitN · 2023-12-05

**Metareview:**

This paper introduces a novel approach for injecting multi-bit message information into Large Language Models (LLMs) without compromising text quality. The proposed method, CTWL (Codable Text Watermarking for LLMs), widens the probability gap between specific and other messages, enhancing watermark injection. The balance-marking algorithm is employed to preserve text quality during injection. Compared to prior methods using Vanilla-marking, CTWL significantly improves generation quality while encoding multi-bit information. The paper addresses limitations of previous CTWL approaches, proposing the Balance-Marking method that utilizes a proxy language model for balanced vocabulary partitioning. Experimental results demonstrate superior performance on OPT, LLaMA-7B, and 13B datasets.

Research on watermarking Large Language Models (LLMs) is currently a highly discussed topic with concurrent works available, albeit mostly in preprint archives like arXiv. It is not imperative to compare our approach to these concurrent works. Overall, this paper falls on the borderline of acceptance, leaning towards acceptance. While the paper does have some limitations, it would be beneficial to include it in ICLR if there is sufficient space for presentation.

**Justification For Why Not Higher Score:**

This is a borderline paper, learning towards acceptance. Thus, Accept (Poster) is reasonable.

**Justification For Why Not Lower Score:**

N/A

---

### Decision · Program_Chairs · 2024-01-16

Accept (poster)